



# Back-calculation of the 2017 Piz Cengalo-Bondo landslide cascade with
# r.avaflow
*Martin Mergili[1,2], Michel Jaboyedoff[3], José Pullarello[3], Shiva P. Pudasaini[4],*
[1] Institute of Applied Geology, University of Natural Resources and Life Sciences (BOKU), Peter-Jordan-Straße 82,
1190 Vienna, Austria
[2] Geomorphological Systems and Risk Research, Department of Geography and Regional Research, University of Vi-
enna, Universitätsstraße 7, 1010 Vienna, Austria
[3] Institute of Earth Sciences, University of Lausanne, Quartier UNIL-Mouline, Bâtiment Géopolis, 1015 Lausanne,
Switzerland
[4] Institute of Geosciences and Meteorology, Geophysics Section, University of Bonn, Meckenheimer Allee 176, 53115
Bonn, Germany
Correspondence to: M. Mergili (martin.mergili@boku.ac.at)
## Abstract
In the morning of 23 August 2017, around 3 million m³ of granitoid rock broke off from the east face of Piz Cengalo,
SE Switzerland. The initial rock slide-rock fall entrained 0.6 million m³ of a glacier and continued as a rock(-ice) ava-
lanche, before evolving into a channelized debris flow that reached the village of Bondo at a distance of 6.5 km after a
couple of minutes. Subsequent debris flow surges followed in the next hours and days. The event resulted in eight
fatalities along its path and severely damaged Bondo. The most likely candidates for the water causing the transfor-
mation of the rock avalanche into a long-runout debris flow are the entrained glacier ice and water originating from
the debris beneath the rock avalanche. In the present work we try to reconstruct conceptually and numerically the
cascade from the initial rock slide-rock fall to the first debris flow surge and thereby consider two scenarios in terms
of qualitative conceptual process models: (i) entrainment of most of the glacier ice by the frontal part of the initial
rock slide-rock fall and/or injection of water from the basal sediments due to sudden rise in pore pressure, leading to a
frontal debris flow, with the rear part largely remaining dry and depositing mid-valley; and (ii) most of the entrained
glacier ice remaining beneath/behind the frontal rock avalanche, and developing into an avalanching flow of ice and
water, part of which overtops and partially entrains the rock avalanche deposit, resulting in a debris flow. Both sce-
narios can be numerically reproduced with the two-phase mass flow model implemented with the simulation software
r.avaflow, based on plausible assumptions of the model parameters. However, these simulation results do not allow to
conclude on which of the two scenarios is the more likely one. Future work will be directed towards the application of
a three-phase flow model (rock, ice, fluid) including phase transitions, in order to better represent the melting of glac-
ier ice, and a more appropriate consideration of deposition of debris flow material along the channel.
Keywords: Debris flow, Entrainment, High-mountain process chain, Rock avalanche, Two-phase flow model,
r.avaflow





## 1 Introduction

Landslides lead to substantial damages to life, property, and infrastructures every year. Whereas initial landslides in hilly terrain have mostly local effects, landslides in high-mountain areas, with elevation differences of thousands of metres over a few kilometres may form the initial points of process chains which, due to their interactions with glacier ice, snow, lakes, or basal material, sometimes evolve into long-runout debris avalanches, debris flows or floods. Such complex landslide events may occur in remote areas, such as the 2012 Alpl rock-snow avalanche in Austria (Preh and Sausgruber, 2015) or the 2012 Santa Cruz multi-lake outburst event in Peru (Mergili et al., 2018a). If they reach inhabited areas, such events lead to major destruction even several kilometres away from the source and have led to major disasters in the past, such as the 1949 Khait rock avalanche-loess flow in Tajikistan (Evans et al., 2009b); the 1962 and 1970 Huascarán rock fall-debris avalanche events in Peru (Evans et al., 2009a; Mergili et al., 2018b); the 2002 Kolka-Karmadon ice-rock avalanche in Russia (Huggel et al., 2005); the 2012 Seti River debris flood in Nepal (Bhandari et al., 2012); or the 2017 Piz Cengalo-Bondo rock avalanche-debris flow event in Switzerland. The initial fall or slide sequences of such process chains are commonly related to a changing cryosphere such as glacial debuttressing, the formation of hanging glaciers, or a changing permafrost regime (Alean, 1985; Haeberli, 1992; Haeberli et al., 1997, 2016; Huggel et al., 2003, 2010, 2012; Noetzli et al., 2006; Gruber and Haeberli, 2007; Harris et al., 2009; Ravanel and Deline, 2011; Krautblatter et al., 2013; Evans and Delaney, 2014; Haeberli and Whiteman, 2014).

Computer models assist risk managers in anticipating the impact areas, energies, and travel times of complex mass flows. They may also support the confirmation or rejection of conceptual models with regard to the physical mechanisms involved in specific cases and thereby contribute to a better understanding of the processes involved. Conventional single-phase flow models, considering a mixture of solid and fluid components (e.g. Voellmy, 1955; Savage and Hutter, 1989; Iverson, 1997; McDougall and Hungr, 2004; Christen et al., 2010), do not serve for such a purpose. Instead, simulations rely on (i) model cascades, changing from one approach to the next at each process boundary (Schneider et al., 2014; Somos-Valenzuela et al., 2016); or (ii) two- or even multi-phase flow models (Pitman and Le, 2005; Pudasaini, 2012; Mergili et al., 2017). Worni et al. (2014) have highlighted the advantages of (ii) for considering also the process interactions and boundaries. Two- or multi-phase flow models separately consider the solid and the fluid phase, but also phase interactions.

The aim of the present work is to learn about our ability to reproduce sophisticated transformation mechanisms involved in complex, cascading landslide processes, with GIS-based numerical models. For this purpose, we apply the computational tool r.avaflow (Mergili et al., 2017), which employs an enhanced version of the Pudasaini (2012) two-phase flow model, to back-calculate the 2017 Piz Cengalo-Bondo landslide cascade in SE Switzerland, which was characterized by the transformation of a rock avalanche to a long-runout debris flow. We consider two scenarios in terms of hypothetic qualitative conceptual models of the physical transformation mechanisms. On this basis, we try to numerically reproduce these scenarios, satisfying the requirements of physical plausibility of the model parameters, and empirical adequacy in terms of correspondence of the results with the documented and inferred impact areas, vol-



umes, velocities, and travel times. Based on the outcomes, we identify the key challenges to be addressed in future
research.
Thereby we rely on the detailed description, documentation, and topographic reconstruction of this recent event. The
event documentation, data used, and the conceptual models are outlined in Section 2. We briefly introduce the simu-
lation framework r.avaflow (Section 3) and explain its parametrization and our simulation strategy (Section 4) before
presenting (Section 5) and discussing (Section 6) the results obtained. Finally, we conclude with the key messages of
the study (Section 7).

## 2  The 2017 Piz Cengalo-Bondo landslide cascade

### 2.1  Piz Cengalo and Val Bondasca

The Val Bondasca is a left tributary valley to the Val Bregaglia in the canton of the Grisons in SE Switzerland (Fig. 1).
The Bondasca stream joins the Mera River at the village of Bondo at 823 m asl. It drains part of the Bregaglia Range,
built up by a mainly granitic intrusive body culminating at 3678 m asl. Piz Cengalo, with a summit elevation of
3368 m asl, is characterized by a steep, intensely fractured NE face which has repeatedly been the scene of landslides,
and which is geomorphologically connected to the Val Bondasca through a steep glacier forefield. The glacier itself has
largely retreated to the cirque beneath the rock wall.
On 27 December 2011, a rock avalanche with a volume of 1.5–2 million m³ developed out of a rock toppling from the
NE face of Piz Cengalo, travelling for a distance of 1.5 km down the Val Bondasca (Haeberli et al., 2013; De Blasio and
Crosta, 2016; Amann et al., 2018). No entrainment of glacier ice was documented for this event, and the deposit did
not connect to the main channel of Val Bondasca, so that no debris flow was recorded and the village of Bondo re-
mained unaffected. As blue ice had been observed directly at the scarp, the role of permafrost for the rock instability
was discussed. An early warning system was installed and later extended (Steinacher et al., 2018). Displacements at the
scarp area were few centimetres per year between 2012 and 2015, and accelerated in the following years. In early Au-
gust 2017, increased rock fall activity and deformation rates alerted the authorities. A major rock fall event occurred
on 21 August 2017 (Amann et al., 2018).

### 2.2  The event of 23 August 2017

At 9:31 am local time on 23 August 2017, a volume of 3.1–3.5 million m³ detached from the NE face of Piz Cengalo, as
indicated by WSL (2017); Amann et al. (2018); and the point cloud we obtained through structure from motion using
pictures taken after the event. Documented by videos and by seismic records (Walter et al., 2018), it evolved into a
rock avalanche which impacted the glacier beneath the rock face and entrained approx. 0.6 million m³ of ice (VAW,
2017; WSL, 2017). Part of the rock avalanche immediately converted into a debris flow which flowed down the Val
Bondasca. It was detected at 9:34 by the debris flow warning system which had been installed near the hamlet of Prä
approx. 1 km upstream from Bondo. According to different sources, the debris flow surge arrived at Bondo between



9:42 (derived from WSL, 2017) and 9:48 (Amt für Wald und Naturgefahren, 2017). The rather low velocity in the low-
er portion of the Val Bondasca is most likely a consequence of the narrow gorge topography, and of the viscous behav-
iour of this first surge. Whereas approx. 540,000 m³ of material were involved, only 50,000 m³ arrived at Bondo im-
mediately (WSL, 2017). The remaining material was partly remobilized by six further debris flow surges recorded dur-
ing the same day, one on 25 August, and one – triggered by rainfall – on 31 August 2017. All nine surges together de-
posited a volume of approx. 500,000–800,000 m³ in the area of Bondo, less than half of which was captured by a reten-
tion basin (Bonanomi and Keiser, 2017).
The total angle of reach of the process chain from the initial release down to the village of Bondo was approx. 18°,
computed from the travel distance of 7.2 km and the vertical drop of approx. 2.3 km. The initial landslide to the ter-
minus of the rock avalanche showed an angle of reach of approx. 28°, derived from the travel distance of 3.3 km and
the vertical drop of 1.8 km. There were eight fatalities, concerning hikers in the Val Bondasca, extensive damages to
buildings and infrastructures, and evacuations for several weeks or even months.

### 112    2.3    Data and conceptual model

Reconstruction of the rock and glacier volumes involved in the event was based on an overlay of a 2011 swisstopo
MNS-Digital Elevation Model (DEM) (contract: swisstopo–DV084371), derived through airborne laser scanning in
2011 and available at a raster cell size of 2 m, and a Digital Surface Model (DSM) obtained through Structure from
Motion (SfM) techniques after the 2017 event. This analysis resulted in a detached rock volume of 3.5 million m³,
which is slightly more than the value of 3.15 million m³ reported by Amann et al. (2018), and an entrained ice volume
of 770,000 m³ (Fig. 4). However, these volumes neglect smaller rock falls before and after the large 2017 event, and
also glacial retreat. The 2011 event took place after the DTM had been acquired, but it released from an area above the
2017 scarp and does therefore not affect the volume reconstruction. Assuming some minor entrainment of the glacier
ice in 2011 and some glacial retreat, we arrive at an entrained ice volume of 600,000 m³, a value which is very well
supported by VAW (2017).
There is still disagreement on the origin of the water having led to the debris flow, particularly to the first surge. Bo-
nanomi and Keiser (2017) clearly mention meltwater from the entrained glacier ice as the main source, whereby much
of the melting is assigned to impact, shearing and frictional heating directly at or after impact, as it is often the situa-
tion in rock-ice avalanches (Pudasaini and Krautblatter, 2014). WSL (2017) has shown, however, that the energy re-
leased was only sufficient to melt approx. half of the glacier ice. Water pockets in the glacier or a stationary water
source along the path might have played an important role (Demmel, 2019). Walter et al. (2019) claim that much of
the glacier ice was crushed, ejected and dispersed (Fig. 3b), whereas water injected into the rock avalanche due to pore
pressure rise in the basal sediments would have played a major role. In any case, the development of a debris flow
from a landslide mass with an overall solid fraction of as high as ~0.85 (considering the water equivalent of the glacier
ice) requires some spatio-temporal differentiation of the water/ice content. We consider two qualitative conceptual
models – or scenarios – possibly explaining such a differentiation:




A. The initial rock slide-rock fall led to massive entrainment, fragmenting and melting of glacier ice, mixing of
rock with some of the entrained ice and the meltwater, and injection of water from the basal sediments into
the rock avalanche mass quickly upon impact due to overload-induced pore pressure rise. As a consequence,
the front of the rock avalanche was characterized by a high content of ice and water, highly mobile, and
therefore escaped as the first debris flow surge, whereas the less mobile rock avalanche behind – still with
some water and ice in it – decelerated and deposited mid-valley. The secondary debris flow surges occurred
mainly due to backwater effects. This scenario largely follows the explanation of Walter et al. (2019) that the
first debris flow surge was triggered at the front of the rock avalanche by overload and pore pressure rise,
whereas the later surges overtopped the rock avalanche deposits, as indicated by the surficial scour patterns.

B. The initial rock slide-rock fall impacted and entrained the glacier. Most of the entrained ice remained beneath
and developed into an avalanching flow of melting ice behind the rock avalanche. The rock avalanche decel-
erated and stopped mid-valley. Part of the avalanching flow overtopped and partly entrained the rock ava-
lanche deposit – leaving behind the scour traces observed in the field – and evolved into the channelized de-
bris flow which arrived at Bondo a couple of minutes later. The secondary debris flow surges started from the
rock avalanche deposit due to melting and infiltration of the remaining ice, and due to backwater effects. This
scenario is similar to the theory developed at the WSL Institute for Snow and Avalanche Research (SLF), who
also did a first simulation of the rock avalanche (WSL, 2017).

Fig. 5 illustrates the conceptual models attempting to explain the key mechanisms involved in the rock avalanche-
debris flow transformation.

## 3   The simulation framework r.avaflow

r.avaflow represents a comprehensive GIS-based open source framework which can be applied for the simulation of
various types of geomorphic mass flows. In contrast to most other mass flow simulation tools, r.avaflow utilizes a gen-
eral two-phase-flow model describing the dynamics of the mixture of solid particles and viscous fluid and the strong
interactions between these phases. It further considers erosion and entrainment of surface material along the flow
path. These features facilitate the simulation of cascading landslide processes such as the 2017 Piz Cengalo-Bondo
event. r.avaflow is outlined in full detail by Mergili et al. (2017). The code, a user manual, and a collection of test da-
tasets are available from Mergili (2019). Only those aspects directly relevant for the present work are described in this
section.
Essentially, the Pudasaini (2012) two-phase flow model is employed for computing the dynamics of mass flows moving
from a defined release area (solid and/or fluid heights are assigned to each raster cell) or release hydrograph (at each
time step, solid and/or fluid heights are added at a given profile, moving at a given cross-profile velocity) down
through a DEM. The spatio-temporal evolution of the flow is approximated through depth-averaged solid and fluid
mass and momentum balance equations (Pudasaini, 2012). This system of equations is solved through the TVD-NOC





Scheme introduced by Nessyahu and Tadmor (1990), adapting an approach presented by Tai et al. (2002) and Wang et
al. (2004). The characteristics of the simulated flow are governed by a set of flow parameters (some of them are shown
in the Tables 1 and 2). Compared to the Pudasaini (2012) model, some extensions have been introduced which include
(i) ambient drag or air resistance (Kattel et al., 2016; Mergili et al., 2017); and (ii) fluid friction, governing the influ-
ence of basal surface roughness on the fluid momentum (Mergili et al., 2018b). Both extensions rely on empirical coef-
ficients, $C_{AD}$ for the ambient drag and $C_{FF}$ for the fluid friction. Further, drag and viscosity are computed according to
enhanced concepts. Most importantly, the internal friction angle $\varphi$ and the basal friction angle $\delta$ of the solid are scaled
with the solid fraction in order to approximate effects of reduced interaction between the solid particles and the basal
surface in fluid-rich flows.
Entrainment is calculated through an empirical model. In contrast to Mergili et al. (2017), where an empirical en-
trainment coefficient is multiplied with the momentum of the flow, here we multiply the entrainment coefficient
$C_E$ (s kg$^{-1}$ m$^{-1}$) with the kinetic energy of the flow:
$$q_{E,s} = C_E |T_s + T_f| \alpha_{s,E}, \quad q_{E,f} = C_E |T_s + T_f| (1 - \alpha_{s,E}), \quad (1)$$

where $q_{E,s}$ and $q_{E,f}$ (m s$^{-1}$) are the solid and fluid entrainment rates, $T_s$ and $T_f$ (J) are the kinetic energies of the solid and
fluid fractions of the flow, and $\alpha_{s,E}$ is the solid fraction of the entrainable material. Solid and fluid flow heights and
momenta, and the change of the basal topography, are updated at each time step (see Mergili et al., 2017 for details).
As r.avaflow operates on the basis of GIS raster cells, its output essentially consists of raster maps –for all time steps
and for the overall maximum – of solid and fluid flow heights, velocities, pressures, kinetic energies, and entrained
heights. In addition, output hydrograph profiles may be defined at which solid and fluid heights, velocities, and dis-
charges are provided at each time step.

## 187   4   Parameterization of r.avaflow

One set of simulations is performed for each of the Scenarios A and B (Fig. 5), considering the process chain from the
release of the rock slide-rock fall to the arrival of the first debris flow surge at Bondo. Neither triggering of the event
nor subsequent surges or distal debris floods beyond Bondo are considered in this study. Equally, the dust cloud associ-
ated to the rock avalanche (WSL, 2017) is not the subject here. Initial sliding of the glacier beneath the rock ava-
lanche, as assumed in Scenario B, cannot directly be modelled. That would require a three-phase model, which is be-
yond the scope here. Instead, release of the glacier ice and meltwater is assumed in a separate simulation after the rock
avalanche has passed over it. We consider this workaround an acceptable approximation of the postulated scenario
(Section 6).
We use the 2011 swisstopo MNS-DEM, corrected for the rock slide-rock fall scarp and the entrained glacier ice by
overlay with the 2017 SfM DSM (Section 2). The maps of release height and maximum entrainable height are derived
from the difference between the 2011 swisstopo DTM and the 2017 SfM DSM (Fig. 4; Section 2). The release mass is





considered completely solid, whereas the entrained glacier is assumed to contain some solid fraction (coarse till). The
glacier ice is assumed to melt immediately on impact and is included in the fluid along with fine till. We note that the
fluid phase does not represent pure water, but a mixture of water and fine particles (Table 2). The fraction of the glaci-
er allowed to be incorporated in the process chain is empirically optimized (Table 3). Based on the same principle, the
maximum depth of entrainment of fluid due to pore pressure overload in Scenario A is set to 25 cm, whereas the max-
imum depth of entrainment of the rock avalanche deposit in Scenario B is set to 1 m.
The study area is divided into six zones A–F (Fig. 6; Table 1). Each of these zones represents an area with particular
surface and flow characteristics, which can be translated into model parameters. Due to the impossibility to directly
measure the key parameters in the field (Mergili et al., 2018a, b), the parameters summarized in Table 1 and Table 2
are the result of an iterative optimization procedure, where multiple simulations with different parameter sets are
performed in order to arrive at one "optimum" simulation for each scenario. It is thereby important to note that we
largely derive one single set of optimized parameters, which is valid for both of the scenarios. Optimization criteria are
(i) the empirical adequacy of the model results, and (ii) the physical plausibility of the parameters. Thereby, the empir-
ical adequacy is quantified through comparison of the results with the documented impact area, the travel times to the
output hydrograph profiles O2, O3, and O4 (Fig. 6), and the reported volumes (Amt für Wald und Naturgefahren,
2017; Bonanomi and Keiser, 2017; WSL, 2017). The physical plausibility of the model parameters is evaluated on the
basis on the parameters suggested by Mergili et al. (2017) and on the findings of Mergili et al. (2018a, b). We note that
the values of the basal friction angle ($\delta$), the ambient drag coefficient ($C_{AD}$), the fluid friction coefficient ($C_{FF}$), and the
entrainment coefficient ($C_E$) are differentiated between and within the zones (Table 1), whereas global values are de-
fined for all the other parameters (Table 2).
Durations of $t = 1800$ s are considered for both scenarios. At this point of time, the first debris flow surge has largely
passed and left the area of interest, except for some remaining tail of fluid material. Only heights ≥0.25 m are taken
into account for the visualization and evaluation of the simulation results. Considering the size of the event, a cell size
of 10 m is considered the best compromise between capturing a sufficient level of detail and ensuring an adequate
computational efficiency, and is therefore applied for all simulations.

## 5   Simulation results

### 5.1   Scenario A – Frontal debris flow surge

Fig. 7 illustrates the distribution of the simulated maximum flow heights, maximum entrained heights, and deposition
area after $t = 1800$ s, when most of the initial debris flow surge has passed the confluence of the Bondasca stream and
the Maira river. The comparison of observed and simulated impact areas results in a critical success index $CSI = 0.568$,
a distance to perfect classification $D2PC = 0.149$, and a factor of conservativeness $FoC = 1.523$. These performance in-
dicators are derived from the confusion matrix of true positives, true negatives, false positives, and false negatives.
They are explained in more detail by Mergili et al. (2018b). Interpreting these values as indicators for a reasonably





good correspondence between simulation and observation in terms of impact area, we now consider the dimension of
time, focussing on the output hydrographs OH1–OH4 (Fig. 8; see Fig. 6 and Fig. 7 for the location of the correspond-
ing hydrograph profiles O1–O4). Most of the rock avalanche passes the profile O1 between $t = 40$ s and $t = 100$ s. OH2
(Fig. 8a; located in the upper portion of Val Bondasca) sets on before $t = 140$ s and quickly reaches its peak, with a vol-
umetric solid ratio of approx. 34% (maximum 900 m³/s of solid and 1,760 m³/s of fluid discharge). Thereafter, this first
surge quickly tails off and then remains at total discharge values below 400 m³/s. The solid flow height, however, re-
mains above 2 m until the end of the simulation, whereas the fluid flow height slowly and steadily tails off. Until
$t = 1800$ s the profile O2 is passed by a total of 183,000 m³ of solid and 252,000 m³ of fluid material (the fluid repre-
senting a mixture of fine mud and water with a density of 1,400 kg m⁻³; see Table 2). The hydrograph profile O3 in
Prä, approx. 1 km upstream of Bondo, is characterized by a surge starting before $t = 260$ s and slowly tailing off after-
wards. Discharge at the hydrograph OH4 (Fig. 8b; O4 is located at the outlet of the canyon to the debris fan of Bondo)
starts before t = 740 s and reaches its peak of solid discharge at $t = 940$ s (89 m³/s). Solid discharge decreases thereafter,
whereas the flow becomes fluid-dominated with a fluid peak of 135 m³/s at $t = 1440$ s. The maximum total flow height
simulated at O4 is 1.25 m. This site is passed by a total of 49,000 m³ of solid and 108,000 m³ of fluid material, according
to the simulation – an overestimate, compared to the documentation (Table 3).
Fig. 9 illustrates the travel time and the frontal velocities of the rock avalanche and the initial debris flow. The initial
surge reaches the hydrograph profile O3 – located 1 km upstream of Bondo – at $t = 260$ s (Fig. 9a; Fig. 8c). This is in
line with the documented arrival of the surge at the nearby monitoring station (Table 3). Also the simulated travel
time to the profile O4 corresponds to the – though uncertain – documentation. The initial rock avalanche is character-
ized by frontal velocities >25 m/s, whereas the debris flow largely moves at 15–25 m/s. Velocities drop below 15 m/s in
the upper part of the Val Bondasca (Zone D), and below 5 m/s in the lower part of the valley (Zone E) (Fig. 9b).
**5.2    Scenario B – Debris flow surge by overtopping and entrainment of rock avalanche**
Fig. 10 illustrates the distribution of the simulated maximum flow heights, maximum entrained heights, and deposi-
tion area after $t = t_0 + 1740$ s, where $t_0$ is the time between the release of the initial rock avalanche and the mobiliza-
tion of the entrained glacier. The simulated impact and deposition areas of the initial rock avalanche are also shown in
Fig. 10. However, we now concentrate to the debris flow, triggered by the entrainment of 150,000 m³ of solid material
from the rock avalanche deposit. Flow heights – as well as the hydrographs presented in Fig. 8c and d and the tem-
poral patterns illustrated in Fig. 11 – only refer to the debris flow developing from the entrained glacier and the en-
trained rock avalanche material. The confusion matrix of observed and simulated impact areas reveals partly different
patterns of performance than for the Scenario A: $CSI = 0.614$; $D2PC = 0.278$; and $FoC = 0.904$. The lower $FoC$ value
and the lower performance in terms of $D2PC$ reflect the missing initial rock avalanche in the simulation results. The
output hydrographs OH2 and OH4 differ from the hydrographs obtained through the Scenario A, but also show some
similarities (Fig. 8c and d). Most of the flow passes through the hydrograph profile O1 between $t = t_0 + 40$ s and
$t_0 + 80$ s, and through O2 between $t = t_0 + 120$ s and $t_0 + 180$ s. The hydrograph OH2 is characterized by a short peak of



2,700 m³/s of solid and 3,400 m³/s of fluid, with a volumetric solid fraction of 0.45 and quickly decreasing discharge
and solid fraction afterwards (Fig. 8c). In contrast to Scenario A, flow heights drop steadily, with values below 2 m
from $t = t_0 + 920$ s onwards. The hydrograph OH3 is characterized by a surge starting before $t = t_0 + 260$ s. Discharge at
the hydrograph OH4 (Fig. 8d) starts at $t = t_0 + 740$ s, and the solid peak of 160 m³/s is simulated approx. at
$t = t_0 + 1080$ s. The delay of the peak of fluid discharge is less pronounced when compared to Scenario A (265 m³/s at
$t = t_0 + 1180$ s). Profile O4 is passed by a total of 53,000 m³ of solid and 143,000 m³ of fluid material. The volumetric
solid fraction drops from above 0.70 at the onset of the hydrograph to almost zero (pure fluid) at the end. The maxi-
mum total flow height at O4 is 3.7 m.
Fig. 11 illustrates the travel times and the frontal velocities of the rock avalanche and the initial debris flow. Assuming
that $t_0$ is in the range of some tens of seconds, the time of arrival of the surge at O3 is in line with the documentation
also for the Scenario B (Fig. 11a; Table 3). The frontal velocity patterns along Val Bondasca are very much in line with
those derived in the Scenario A (Fig. 11b). However, the scenarios differ among themselves in terms of the more pro-
nounced, but shorter peaks of the hydrographs in Scenario B (Fig. 8). This pattern is a consequence of the more sharp-
ly defined debris flow surge. In Scenario A, the front of the rock avalanche deposit constantly "leaks" into Val Bon-
dasca, providing supply for the debris flow also at later stages. In Scenario B, entrainment of the rock avalanche depos-
it occurs relatively quickly, without material supply afterwards. This type of behaviour is strongly coupled to the value
of $C_E$ and the allowed height of entrainment chosen for the rock avalanche deposit.
## 6 Discussion
Our simulation results reveal a reasonable degree of empirical adequacy and physical plausibility with regard to most
of the reference observations. Having said that, we have also identified some important limitations which are now
discussed in more detail. First of all, we are not able to decide on the more realistic of the two Scenarios A and B. In
general, the melting and mobilization of glacier ice upon rock slide-rock fall impact is hard to quantify from straight-
forward calculations of energy transformation, as Huggel et al. (2005) have demonstrated on the example of the 2002
Kolka-Karmadon event. In the present work, the assumed amount of melting (approx. half of the glacier ice) leading to
the empirically most adequate results corresponds well to the findings of WSL (2017), indicating a reasonable degree
of plausibility. It remains equally difficult to quantify the amount of water injected into the rock avalanche by over-
load of the sediments and the resulting pore pressure rise.
We note that with the approach chosen we are not able (i) to adequately simulate the transition from solid to fluid
material; and (ii) to consider rock and ice separately with different material properties, which would require a three-
phase model, not within the scope here. Therefore, entrained ice is considered viscous fluid from the beginning. A
physically better founded representation of the initial phase of the event would require an extension of the flow model
employed. Such an extension could build on the rock-ice avalanche model introduced by Pudasaini and Krautblatter
(2014). Also the vertical patterns of the situation illustrated in Fig. 4 cannot be modelled with the present approach,





which (i) does not consider melting of ice; and (ii) only allows one entrainable layer at each pixel. The assumption of
fluid behaviour of glacier ice therefore represents a necessary simplification which is supported by observations
(Fig. 3b), but neglects the likely presence of remaining ice in the basal part of the eroded glacier, which melted later
and so contributed to the successive debris flow surges.
The initial rock slide-rock fall and the rock avalanche are simulated in a plausible way, at least with regard to the dep-
osition area. Whereas the simulated deposition area is clearly defined in Scenario B, this is to a lesser extent the case in
Scenario A, where the front of the rock avalanche directly transforms into a debris flow. Both scenarios seem to over-
estimate the time between release and deposition, compared to the seismic signals recorded – an issue also reported by
WSL (2017) for their simulation. We observe a relatively gradual deceleration of the simulated avalanche, without
clearly defined stopping and note that also in the Scenario B, there is some diffusion after the considered time of 120 s,
so that the definition of the simulated deposit is somehow arbitrary. The elaboration of well-suited stopping criteria,
going beyond the very simple approach introduced by Mergili et al. (2017), remains a task for the future. However, as
the rock avalanche has already been successfully back-calculated by WSL (2017), we focus on the first debris flow
surge: the simulation input is optimized towards the back-calculation of the debris flow volumes entering the valley at
the hydrograph profile O2 (Table 3). The travel times to the hydrograph profiles O3 and O4 are reproduced in a plau-
sible way in both scenarios, and so are the impact areas (Figs. 7 and 10). Exceedance of the lateral limits in the lower
zones is attributed to an overestimate of the debris flow volumes there, and to numerical issues related to the narrow
gorge. The solid ratio of the debris flow in the simulations appears realistic, ranging around 45% in the early stage of
the debris flow, and around 30% in the final stage. This means that solid material tends to stop in the transit area ra-
ther than fluid material, as it can be expected. Nevertheless, the correct simulation of the deposition of debris flow
material along Val Bondasca remains a major challenge (Table 3). Even though a considerable amount of effort was put
in reproducing the much lower volumes reported in the vicinity of O4, the simulations result in an overestimate of the
volumes passing through this hydrograph profile. This is most likely a consequence of the failure of r.avaflow to ade-
quately reproduce the deposition pattern in the zones D and E. Whereas some material remains there at the end of the
simulation, and even more material is lost due to numerical diffusion, more work is necessary to appropriately under-
stand the mechanisms of deposition in viscous debris flows (Pudasaini and Fischer, 2016b). Part of the discrepancy,
however, might be explained by the fact that part of the fluid material – which does not only consist of pure water,
but of a mixture of water and fine mud – left the area of interest in downstream direction and was therefore not in-
cluded in the reference measurements.
The simulation results are strongly influenced by the initial conditions and the model parameters. Parameterization of
both scenarios is complex and highly uncertain, particularly in terms of optimizing the volumes of entrained till and
glacial meltwater, and injected pore water. In general, the parameter sets optimized to yield empirically adequate re-
sults are physically plausible, in contrast to Mergili et al. (2018b) who had to set the basal friction angle in a certain
zone to a negligible value in order to reproduce the observed overtopping of a more than 100 m high ridge (1970
Huascarán landslide). In contrast, reproducing the travel times to O4 in the present study requires the assumption of a



low mobility of the flow in Zone E. This is achieved by increasing the friction (Table 1), accounting for the narrow
flow channel, i.e. the interaction of the flow with the channel walls, which is not directly accounted for in r.avaflow.
Still, the high values of $\delta$ given in Table 1 are not directly applied, as they scale with the fluid content. This type of
weighting has to be further scrutinized. We emphasize that also reasonable parameter sets are not necessarily true, as
the large number of parameters involved (Tables 1 and 2) creates a lot of space for equifinality issues (Beven et al.,

339  1996).

We have further shown that the classical evaluation of empirical adequacy, by comparing observed and simulated
impact areas, is not enough in the case of complex mass flows: travel times, hydrographs, and volumes involved can
provide important insight in addition to the classical quantitative performance indicators used, for example, in land-
slide susceptibility modelling (Formetta et al., 2015). Further, the delineation of the observed impact area is uncertain
as the boundary of the event is not clearly defined particularly in Zone C.

## 7  Conclusions

Both of the investigated Scenarios A and B lead to empirically reasonably adequate results, when back calculated with
r.avaflow using physically plausible model parameters. Based on the simulations performed in the present study, final
conclusions on the more likely of the mechanisms sketched in Fig. 5 can therefore not be drawn purely based on the
simulations. The observed jet of glacial meltwater (Fig. 3b) points towards Scenario A. The observed scouring of the
rock avalanche deposit, in contrast, rather points towards Scenario B, but could also be associated to subsequent debris
flow surges. Open questions include at least (i) the interaction between the initial rock slide-rock fall and the glacier;
(ii) flow transformations in the lower portion of Zone C (Fig. 6), leading to the first debris flow surge; and (iii) the
mechanisms of deposition of 90% of the debris flow material along the flow channel in the Val Bondasca. Further re-
search is therefore urgently needed to shed more light on this extraordinary landslide cascade in the Swiss Alps. In
addition, improved simulation concepts are needed to better capture the dynamics of complex landslides in glacierized
environments: such would particularly have to include three-phase models, where ice – and melting of ice – are con-
sidered in a more explicit way.

## Acknowledgements

Shiva P. Pudasaini gratefully thanks the Herbette Foundation for providing financial support for his sabbatical visit to
the University of Lausanne, Switzerland in the period April–June 2018, where this contribution was triggered. Simi-
larly, this work has been financially supported by the German Research Foundation (DFG) through the research pro-
ject PU 386/5-1: "A novel and unified solution to multi-phase mass flows". It strongly builds on the outcomes of the
international cooperation project "A GIS simulation model for avalanche and debris flows (avaflow)" supported by the
German Research Foundation (DFG, project number PU 386/3-1) and the Austrian Science Fund (FWF, project num-
ber I 1600-N30).





We are further grateful to Sophia Demmel and Florian Amann for valuable discussions and to Matthias Benedikt for
comprehensive technical assistance.

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





## Tables

Table 1. Descriptions and optimized parameter values for each of the zones A–F (Fig. 6). The names of the model parameters are given in the text and in Table 2. The values provided in Table 2 are assigned to those parameters not shown. (A) and (B) refer to the corresponding scenarios.

| Zone | Description | Model parameters | Initial conditions |
|---|---|---|---|
| A | Rock zone – NE face of Piz Cengalo with rock slide-rock fall release area | $\delta = 20°$ (A)[1] <br> $\delta = 15°$ (B)[2] <br> $C_{AD} = 0.2$ | Release volume: <br> 3.46 million m³, 100 % solid[3] |
| B | Glacier zone – Cirque glacier beneath zone A, entrainment of glacier ice[1] | $\delta = 20°$ (A) <br> $\delta = 15°$ (B) <br> $C_E = 10^{-6.5}$ | Entrainment of glacier ice and till (Table 3)[4] |
| C | Slope zone – steep, partly debris-covered glacier forefield leading down to the Val Bondasca | $\delta = 20°$ (A) <br> $\delta = 15°$ (B) <br> $C_E = 10^{-6.5}$ (A) <br> $C_E = 10^{-8.0}$ (B) | Entrainment of injected water in Scenario A <br><br> Entrainment of rock avalanche deposit in Scenario B |
| D | Upper Val Bondasca zone – clearly defined flow channel becoming narrower in downstream direction | $\delta = 20\text{-}45°$ | No entrainment allowed, increasing friction |
| E | Lower Val Bondasca zone – narrow gorge | $\delta = 45°$ <br> $C_{FF} = 0.5$ | No entrainment allowed, high friction due to lateral confinement |
| F | Bondo zone – deposition of the debris flow on the cone of Bondo | $\delta = 20°$ | No entrainment allowed |

[1] Note that in all zones and in both of the scenarios A and B, $\delta$ is assumed to scale linearly with the solid fraction. This means that the values given only apply in case of 100% solid.

[2] This only applies to the initial landslide, which is assumed completely dry in Scenario B. Due to the scaling of $\delta$ with the solid fraction, a lower basal friction is required to obtain results similar to Scenario A, where the initial landslide contains some fluid. The same values of $\delta$ as for Scenario A are applied for the debris flow in Scenario B throughout all zones.

[3] This volume is derived from our own reconstruction (Fig. 4). In contrast, WSL (2017) gives 3.1 million m³, and Amann et al. (2018) 3.15 million m³.

[4] In Scenario B, the glacier is not directly entrained, but instead released behind the rock avalanche. In both scenarios, ice is considered to melt immediately on impact and included in the viscous fluid fraction. See text for more detailed explanations.





Table 2. Model parameters used for the simulations.

| Symbol | Parameter | Unit | Value |
|---|---|---|---|
| $\rho_S$ | Solid material density (grain density) | kg m$^{-3}$ | 2,700 |
| $\rho_F$ | Fluid material density | kg m$^{-3}$ | 1,400[1] |
| $\varphi$ | Internal friction angle | Degree | 27[2] |
| $\delta$ | Basal friction angle | Degree | Table 1 |
| $v$ | Kinematic viscosity of the fluid | m² s$^{-1}$ | 10 |
| $\tau_Y$ | Yield strength of the fluid | Pa | 10 |
| $C_{AD}$ | Ambient drag coefficient | – | 0.04 (exceptions in Table 1) |
| $C_{FF}$ | Fluid friction coefficient | | 0.0 (exceptions in Table 1) |
| $C_E$ | Entrainment coefficient | – | Table 1 |

[1] Fluid is here considered as a mixture of water and fine particles. This explains the higher density, compared to pure
water.
[2] The internal friction angle $\varphi$ always has to be larger than or equal to the basal friction angle $\delta$. Therefore, in case of
$\delta > \varphi$, $\varphi$ is increased accordingly.





Table 3. Selected output parameters of the simulations for the Scenarios A and B compared to the observed or docu-
mented parameter values. S = solid; F = fluid; fractions are expressed in terms of volume; $t_0$ = time from the initial re-
lease to the release of the first debris flow surge. Reference values are extracted from Amt für Wald und Naturgefah-
ren (2017a), Bonanomi and Keiser (2017), and WSL (2017). *** = empirically adequate; ** = empirically partly adequate;
* = empirically inadequate.

| Parameter | Documenta-tion/Observation | Scenario A | Scenario B |
|---|---|---|---|
| Entrained ice (m³) | 600,000[1] | – | – |
| Entrained S (m³) | – | 60,000 | 60,000[2] |
| Entrained F (m³) | – | 300,000 | 240,000 |
| Duration of initial landslide | 60–90[3] | 100–120** | 100–120** |
| Travel time to O2 (s) | 90–120[4] | 140** | $t_0$+140** |
| Travel time to O3 (s) | 210–300[5] | 260*** | $t_0$+260*** |
| Travel time to O4 (s) | 630–1020[6] | 740*** | $t_0$+740*** |
| Debris flow volume at O2 (m³) | 540,000 | 440,000*** (42% S***) | 400,000*** (45% S***) |
| Debris flow volume at O4 (m³) | 50,000 | 160,000* (31% S***) | 200,000* (27% S***) |

[1] Not all the material entrained from the glacier was relevant for the first debris flow surge (Fig. 5), therefore lower
volumes of entrained S (coarse till, in Scenario B also rock avalanche deposit) and F (molten ice and fine till, in Scenar-
io A also pore water) yield the empirically most adequate results. The F volumes originating from the glacier in the
simulations represent approx. half of the water equivalent of the entrained ice, corresponding well to the findings of
WSL (2017).
[2] This value does not include the 140,000 m³ of solid material remobilized through entrainment from the rock ava-
lanche deposit in Scenario B.
[3] WSL (2017) states that the rock avalanche came to rest approx. 60 s after release, whereas the seismic signals ceased
90 s after release.
[4] A certain time (here, we assume a maximum of 30 s) has to be allowed for the initial debris flow surge to reach O2,
located slightly downstream of the front of the rock avalanche deposit.
[5] WSL (2017) gives a travel time of 3.5 minutes to Prä, roughly corresponding to the location of O3. It remains unclear
whether this number refers to the release of the initial rock slide-rock fall or (more likely) to the start of the first de-
bris flow surge. Bonanomi and Keiser (2017) give a travel time of roughly four minutes between the initial release and
the arrival of the first surge at the sensor of Prä.
[6] Amt für Wald und Naturgefahren (2017) gives a time span of 17 minutes between the release of the initial rock
slide-rock fall and the arrival of the first debris flow surge at the "bridge" in Bondo. However, it is not indicated to
which bridge this number refers. WSL (2017), in contrast, give a travel time of 7–8 minutes from Prä to the "old
bridge" in Bondo, which, in sum, results in a shorter total travel time as indicated in Amt für Wald und Naturgefahren
(2017). Depending on the bridge, the reference location for these numbers might be downstream from O4.


**Figures**

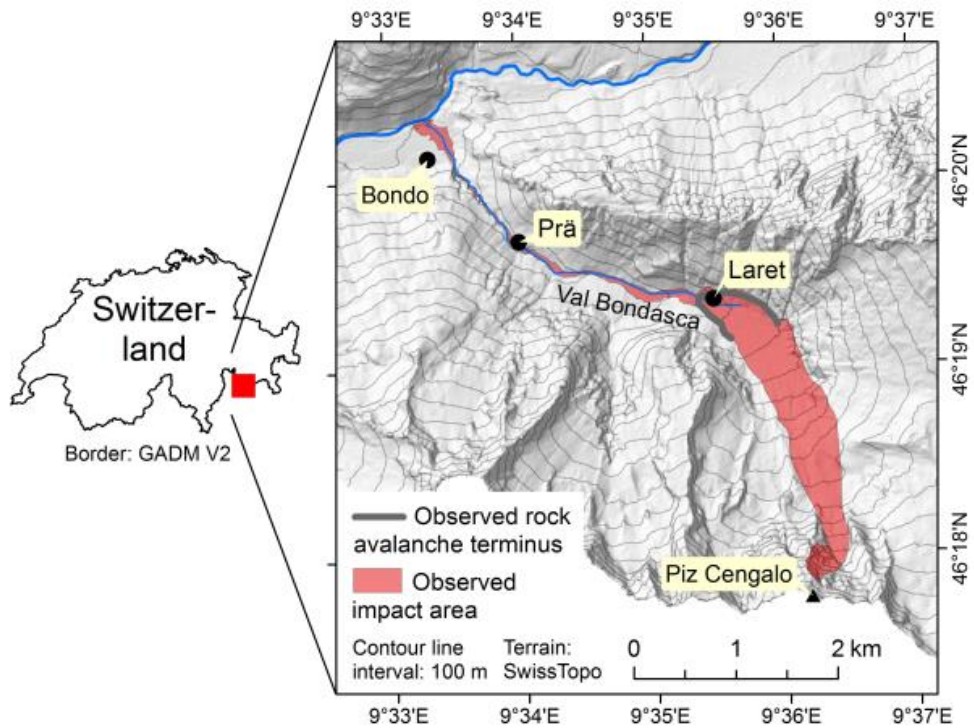

Figure 1. Study area with the impact area of the 2017 Piz Cengalo-Bondo landslide cascade. The observed rock ava-
lanche terminus was derived from WSL (2017).



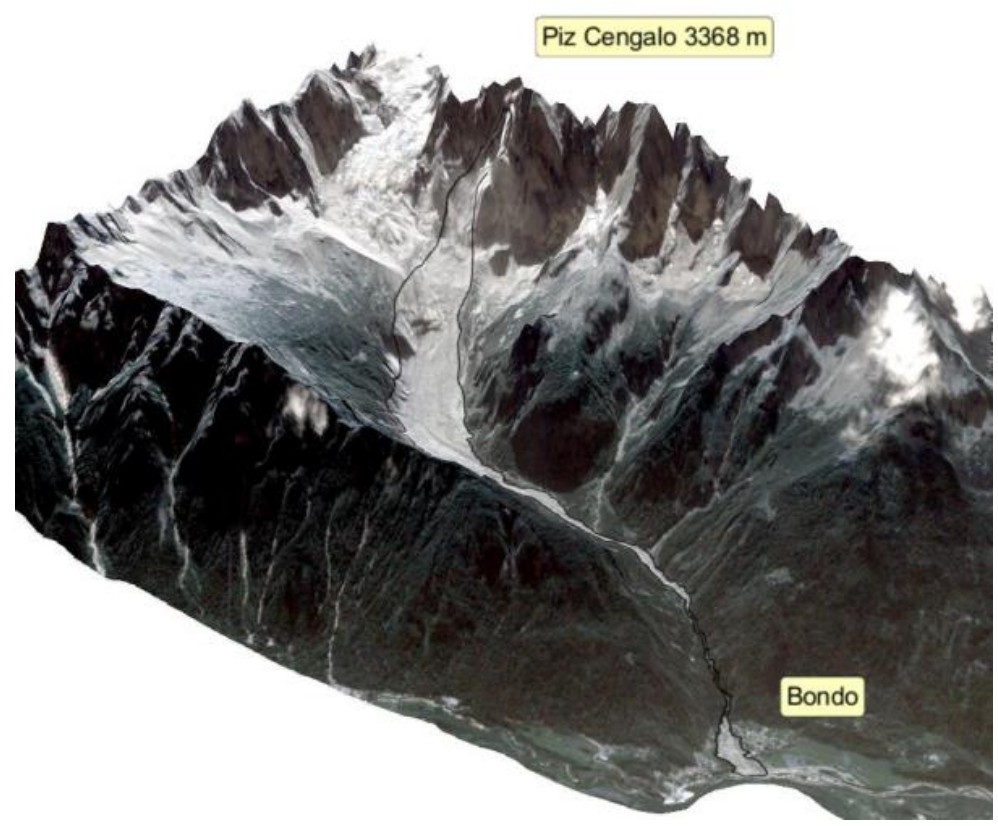

Figure 2. Oblique view of the impact area of the event, orthophoto draped over the 2011 DTM. Data sources: swis-
stopo.


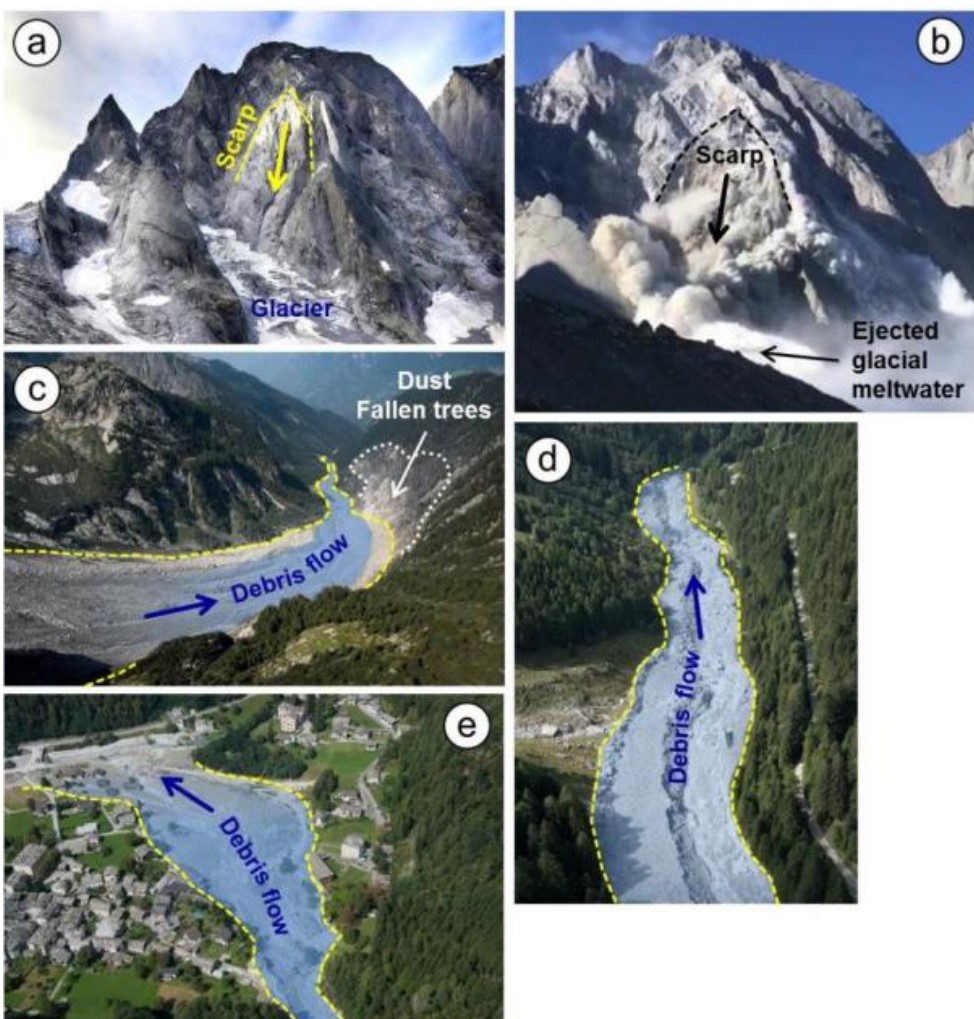

Figure 3. The 2017 Piz Cengalo-Bondo landslide cascade. (a) Scarp area on 20 September 2014. (b) Scarp area on
23 September 2017 at 9:30, 20 s after release, frame of a video taken from the Capanna di Sciora. Note the fountain of
water and/or crushed ice at the front of the avalanche, most likely representing meltwater from the impacted glacier.
(c) Upper part of the Val Bondasca, where the channelized debris flow developed. Note the zone of dust and pressure-
induced damages to trees on the right side of the valley. (d) Traces of the debris flows in the Val Bondasca. (e) The
debris cone of Bondo after the event. Image sources: Daniele Porro (a), Diego Salasc (b), VBS swisstopo Flugdienst (c)–
(e).



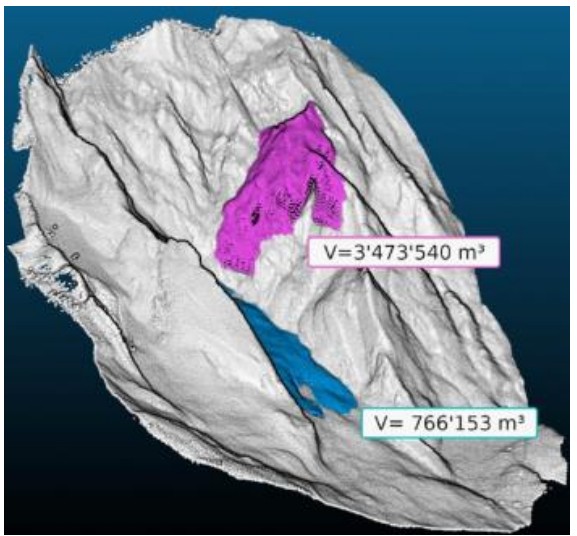

Figure 4. Reconstruction of the released rock volume and the entrained glacier volume in the 2017 Piz Cengalo-Bondo
landslide cascade. Note that the glacier volume shown is neither corrected for entrainment related to the 2011 event,
nor for glacier retreat in the period 2011–2017.





Figure 5. Qualitative conceptual models of the rock avalanche-debris flow transformation. (a) Scenario A; (b) Scenario
B. See text for the detailed description of the two scenarios.


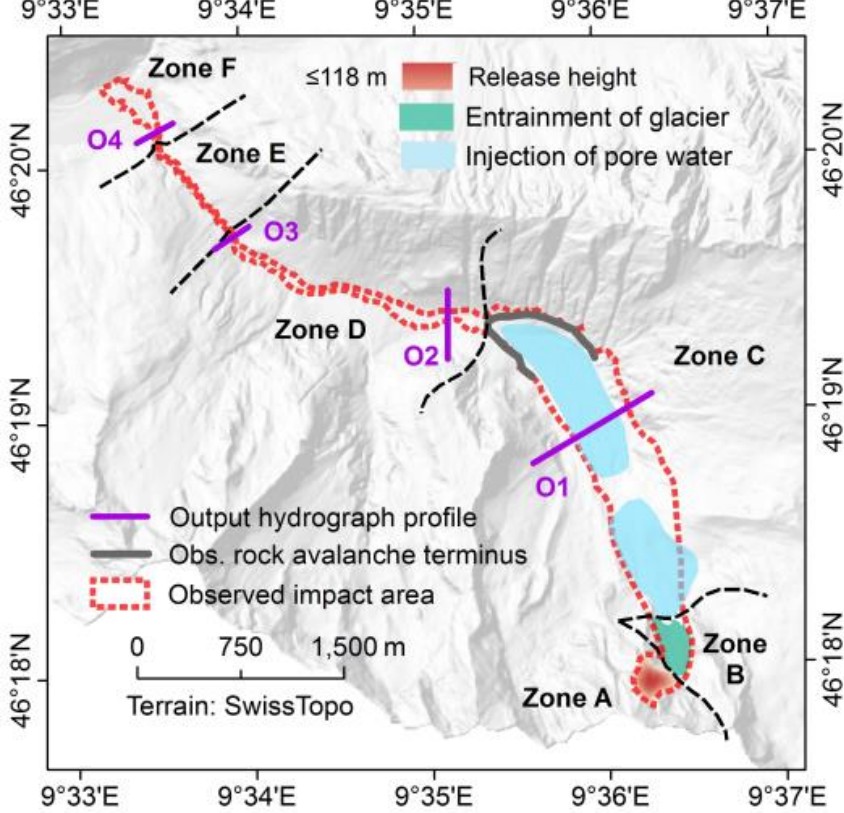

Figure 6. Overview of the heights and entrainment areas as well as the zonation performed as the basis for the simula-
tion with r.avaflow. Injection of pore water only applies to the Scenario A. The zones A–F represent areas with largely
homogeneous surface characteristics. The characteristics of the zones and the model parameters associated to each
zone are summarized in Table 1. O1–O4 represent the output hydrograph profiles. The observed rock avalanche ter-
minus was derived from WSL (2017).

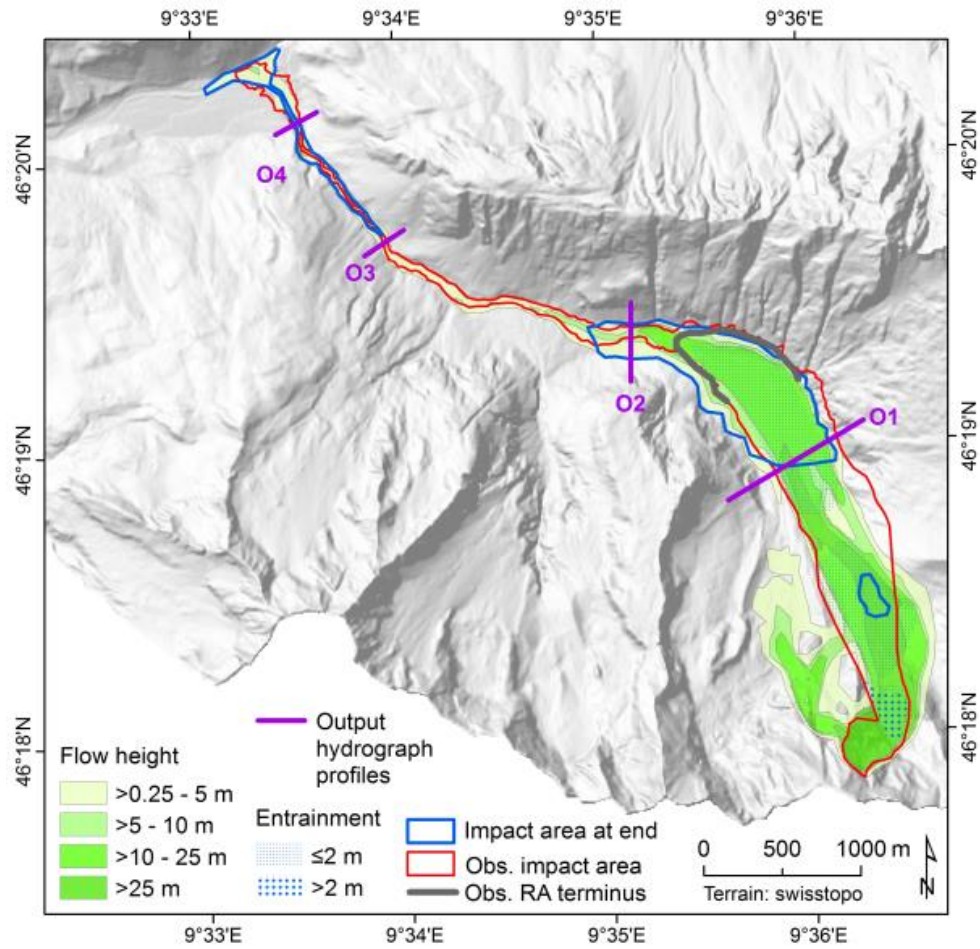

Figure 7. Maximum flow height and entrainment derived for Scenario A. RA = rock avalanche; the observed RA ter-
minus was derived from WSL (2017).

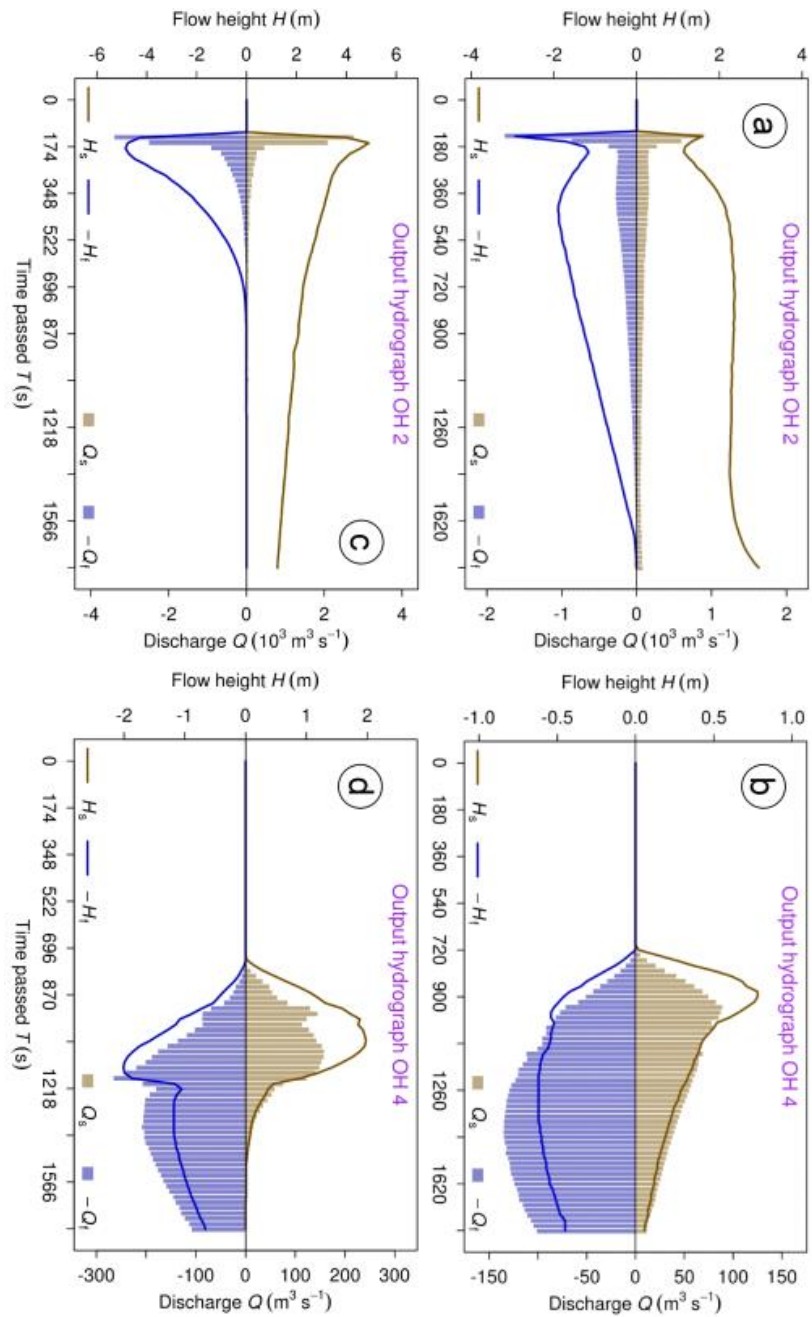

Figure 8. Output hydrographs OH2 and OH4 derived for the scenarios A and B. (a) OH2 for Scenario A. (b) OH4 for
Scenario A. (c) OH2 for Scenario B. (d) OH4 for Scenario B. See Fig. 6 and Fig. 7 for the locations of the hydrograph
profiles O2 and O4. $H_s$ = solid flow height; $H_f$ = fluid flow height; $Q_s$ = solid discharge; $Q_f$ = fluid discharge.

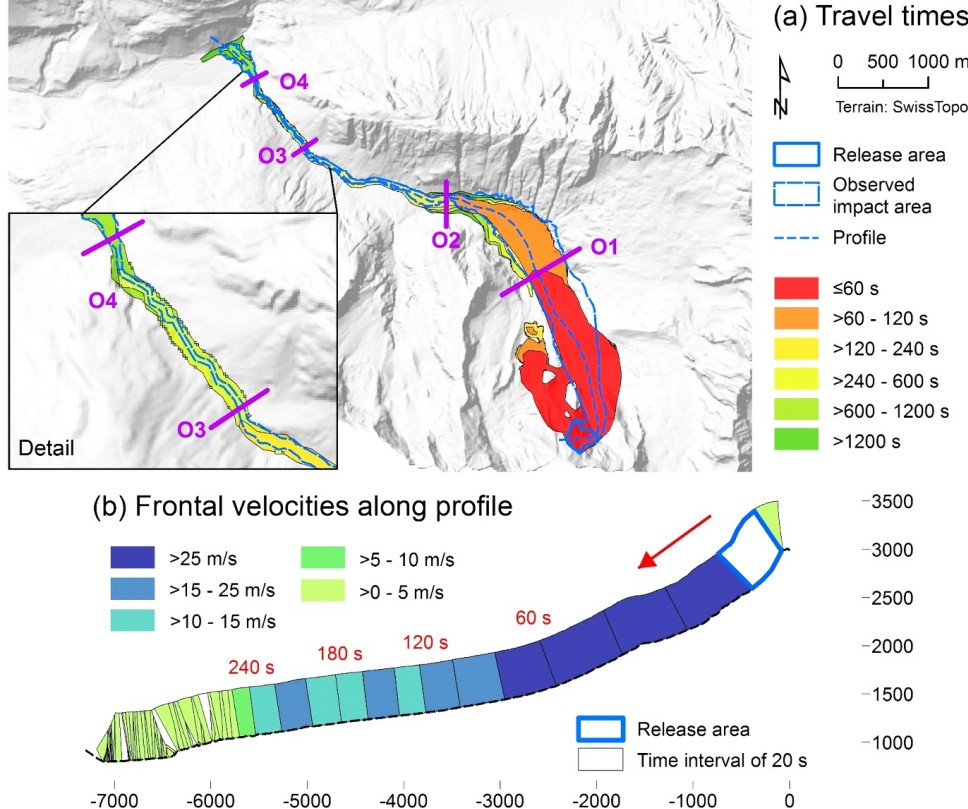

Figure 9. Spatio-temporal evolution and velocities of the event obtained for Scenario A. (a) Travel times, starting from
the release of the initial rock slide-rock fall. (b) Frontal velocities along the flow path, shown in steps of 20 s. Note that
the height of the velocity graph does not scale with flow height. White areas indicate that there is no clear flow path.




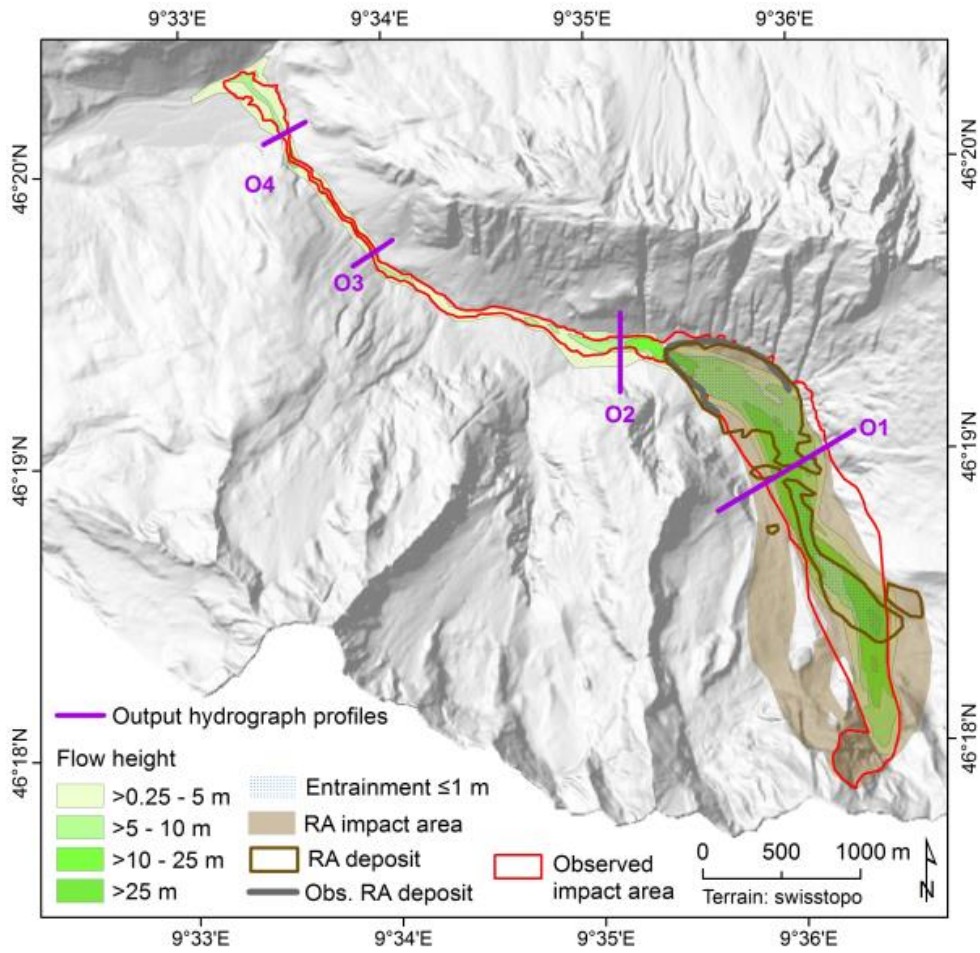

Figure 10. Maximum flow height and entrainment derived for Scenario B. RA = rock avalanche; the observed RA terminus was derived from WSL (2017).

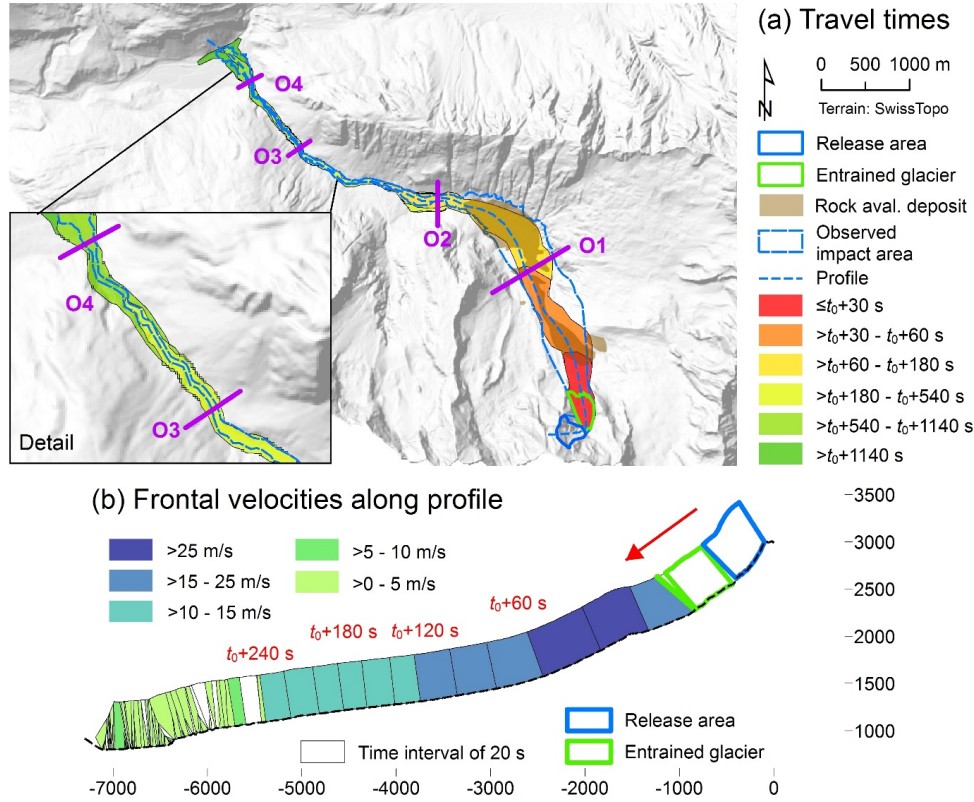

Figure 11. Spatio-temporal evolution and velocities of the event obtained for Scenario B. (a) Travel times, starting from the release of the initial rock slide-rock fall. Thereby $t_0$ (s) is the time between the release of the rock slide-rock fall and the mobilization of the entrained glacier. (b) Frontal velocities along the flow path, shown in steps of 20 s. Note that the height of the velocity graph does not scale with flow height. White areas indicate that there is no clear flow path.