# Peer review of "Back-calculation of the 2017 Piz Cengalo-Bondo landslide cascade with # r.avaflow"

_Natural Hazards and Earth System Sciences, 2019_

## Referee Comment (RC1) · Anonymous Referee #1 · 9 Aug 2019

The authors present an application of the model r.avaflow to the back-calculation of a complex landslide occurred in Switzerland. The case-study is indeed interesting and the scientific question about the two scenario is stimulating (I also really like fig.5). However, as you stated yourself, the investigation of the process through a two-phase numerical model did not allow to discern between the two scenario. So what is really the "take home message" of your work?

I do understand that negative results are results but in a way you do not really present them as such. For example, most of your introduction praise the capabilities of two phase depth averaged model in "support the confirmation or rejection of conceptual models" stating the intrinsic epistemological potentiality with respect to one phase models. However, you then proceed with your modelling, that anyhow requires calibration

and the selection of vague "physical plausible" parameters, that has numerical issues that constrain you to use "physically implausible" parameters and that do not perform well in the reconstruction of the actual phenomenon. So rather than titling your paper as "back calculation of the 2017..." I would suggest to switch it to something such as "challenges and open issues regarding the modelling of the 2017...".

These are my other comments regarding your paper

BROAD COMMENTS

1) Optimization and equifinality: The entrainment in your code is calculated with a calibrated coefficient and based on a depth averaged kinetic energy. You defined 6 zones with different friction angles and other calibrated coefficient. How did you suppose it could lead to a selection to confirm of reject a conceptual model (l 51) when, as you stated yourself in the end, there is an obvious problem of equifinality? This issue is common in back analysis, especially when several parameters are involved in the calibration process. To try to give some "physical plausibility" to the whole parametrization of the backward calculation it is important to:

i) define a straightforward and explicit optimization method

ii) use parameters that are somewhat geotechnically believable with respect to the observed phenomena

iii) provide a clear geomorphological/mechanical reason for each zonation – otherwise of course the more are the zones in which the parameters may be changed, the more the equifinality issue arises.

In my opinion point i) is lacking in your paper. The metrics you use are not so straightforward, especially if you need to jump between two papers to reach their definition and reason of being (is it really Mergili 2018b the best paper to refer to or is it better to go directly to Formetta 2015 and Mergili et al., 2017?). Please devote a paragraph to the interpretation of these metrics rather than cut the discussion off with "indicators of a

reasonably good corrispondance". And what is reasonably anyway? Please also show in the picture a zoom of the deposition pattern (modelled and observed) in the alluvial fan. Point ii) is also important. Finding a 45° angle of friction in the E zone is rather "physically un-plausible" as it is, especially when your solid fraction decreases. You discuss this too briefly in the end of the discussion chapter. You have to explain better what is the issue with the code, is the 10 m sampling? is a projection issue related to the conversion of the coordinates? This should be better discussed, also in the light of the new titling of the manuscript. The zoning (point iii) should be more extensively discussed, the definition that it is found in table 1 is too synthetic. For each zone and limit there must be a defined reason to be it that way.

2) Mass balance: in 220 you write that "only heights <0.25 m are taken into account for the visualization and evaluation of the simulation results". That's ok for the visualization part but what about mass balance? how much impact do have diffusion effects in your model? how much material you discard when you filter at 0.25 m?

SPECIFIC COMMENTS

l 47-49: as a matter of taste I do not think that putting 14 references after a sentence contributes much to the clarity and readability of a paper and to the whole general usefulness for supporting a scientific discourse (that should be the main reason for inserting citations in an introduction)

l 88: insert a couple of words to explain how these displacements were monitored

l 94 and following: check that each acronym has its own definition the first time they appear in the text. Moreover the VAW and WSL reports are written in German so it is not easy to extract the required information. Please if you refer to these works add a sentence summarizing the useful findings.

l 100-102: insert data about the average steepness of the tract. In fact in the whole paper little information about the local heights and steepness are inferable. In Fig 1 the

contour lines labels are missing and in the following maps the contour lines are missing altogether. I would suggest the authors to add the labels in fig. 1 and to insert a table or a figure with the average steepness of the channel profile in each of the 6 zones.

l 109: is it possible to talk about rock avalanche with an angle of reach of 28° and no brandung? refer to Nicoletti 1991, Corominas 1996 and Hungr 2005 and discuss

l 116: did you filtered the errors in the volume estimation? If yes, how?

Fig 8 – please put hydrograph a and b to the left

---

## Referee Comment (RC2) · Brian McArdell (Referee) · 25 Sep 2019

This manuscript describes the application of r.avaflow to a two-process sediment cascade involving a rock avalanche and subsequent debris flows. While the manuscript is interesting, clear and concise and is certainly of interest to users of r.avaflow, the manuscript has the character of a case study and the application of the model and concepts to other field sites is limited by the fact that the model must be calibrated for every new application. This suggests that r.avaflow is perhaps only useful for post-event analysis and not for predictive hazard analyses. r.avaflow is not the only model capable of describing such coupled processes, but readers may well get this impression after reading the manuscript. At a minimum, a discussion of other models and how they also described similar process transitions would be helpful for the reader,

e.g. Iverson et al.'s work on runout modelling of the Oso landslide. The manuscript requires some new literature sections (other models), some new discussion sections (shortcomings and advantages of both models), and a few relatively minor clarifications (below) before it can be published. Some minor points are listed below.

General Comments

1. Is it accurate to call r.avaflow a GIS-based model? Certainly the user interface relies extensively on a GIS interface, but I was under the impression that the model is based on a numerical solution of the shallow-water equations for granular flows, e.g. the flow model written mainly by Shiva Pudasaini. 2. While it is certainly efficient to refer to previous publications about the model and to not re-state the equations used in the model, I think that some description of the equations solved in the model would be appropriate for the reader. Otherwise, it is impossible to understand the manuscript without referring to the paper where the model is described. Does it include terms to describe the influence of curvature on the solution? 3. Section 5: Please define CSI, D2PC, and FoC. Interested readers should at least be able to see how they are defined without having to refer to the Mergili et al., 2018b paper. This is only a few sentences and it would save interested readers a fair amount of work. 4. Lines 211, 284, 290, 340, and possibly elsewhere: I have no idea what you mean by the term "empirical adequacy" which sounds like it could include anything that is physically possible (instead of physically plausible). Please use a different expression. 5. Lines 284—292: I've heard similar statements by Prof. Florian Amann and I believe it was also mentioned by S. Demmel in her masters thesis, or in the WSL 2017 report. I have seen this idea presented this several times at different conferences, so it is unfair to not at least mention that it is not your hypothesis. 6. Others (e.g. Iverson & George) have developed similar two-phase models that are capable of describing process chains. This paper gives a false impression that r.avaflow is the only model capable of describing the runout of coupled processes. Please include a few paragraphs describing and/or comparing and contrasting your model with previous ones.

Specific comments

Lines 85–86: The statement that the deposit did not connect to the main channel of Val Bondasca is incorrect. The 2011 deposit certainly covered the uppermost part of the Val Bondasca and therefore is connected hydrologically and geomorphologically. The main torrent channel after the 2011 event went over and eroded the deposits of the 2011 rock avalanche deposit. Please correct this error, or more precisely state how it was "not connected". This is clearly evident in publications of others, e.g. Frank et al. (2019) which illustrates the triggering of debris flows in Val Bondasca following the 2011 rock avalanche. Frank, F., Huggel, C., McArdell, B. W., & Vieli, A. (2019). Landslides and increased debris-flow activity: a systematic comparison of six catchments in Switzerland. Earth Surface Processes and Landforms, 44(3), 699-712. https://doi.org/10.1002/esp.4524.

Lines 102–103. These data are almost certainly from the Canton of Grisons and not from the WSL, although they were certainly repeated in the WSL 2017 report.

Lines 129–130. This statement about the pore-water pressure rise is somewhat speculative and should be treated as a hypothesis and not as a fact. Rock avalanche deposits (e.g. the 2011 deposits) are often described as having relatively low porosity, and therefore one could, possibly, also expect dilation upon compression which would cause a drop in pore-water pressure and not a rise. The river-bed sediments, and possibly the moraine deposits, upon which the 2017 rock avalanche flowed would be more reasonable candidates for the mechanism you describe here.

Lines 221-222: Did you try other grid resolutions? This resolution, based on my own experience using CFD models (also with a GIS user interface) indicates to me that the grid size is fine for the rock avalanche runout modelling but too coarse for the debris-flow runout modelling.

Line 237: What do you mean by total discharge? The sum of the discharge of all of the phases, or the sum of different flow paths comprising the flow. Or do you mean

maximum discharge or the maximum discharge sum of all of the phases? From the text is appears to be a peak discharge value and not just a total value.

Line 346: A brief re-statement of the Scenarios would be reasonable here for the reader.

Table 1: It's confusing to have Zones A, B, and so on, as well as scenario sets A & B. perhaps you could use lower-case letters for scenarios a and b, or use something like SA or SB to make it completely unambiguous.

Figure 5 is a nice graphical depiction of the two scenarios.

---

## Author Comment (AC2) · 4 Nov 2019

Please see supplement.

Please also note the supplement to this comment:
https://www.nat-hazards-earth-syst-sci-discuss.net/nhess-2019-204/nhess-2019-204-AC2-supplement.pdf
* * *

---

## Author Response (AR1)

**Back-calculation of the 2017 Piz Cengalo-Bondo landslide cascade with r.avaflow**

*Martin Mergili, Michel Jaboyedoff, José Pullarello, Shiva P. Pudasaini*

**Response to the comments of Referee #1**

We would like to thank the reviewer for the constructive remarks. Below, we address each comment in full detail. Our response is written in blue colour. Changes in the manuscript are highlighted in yellow colour.

The authors present an application of the model r.avaflow to the back-calculation of a complex landslide occurred in Switzerland. The case-study is indeed interesting and the scientific question about the two scenario is stimulating (I also really like fig.5). However, as you stated yourself, the investigation of the process through a two-phase numerical model did not allow to discern between the two scenario. So what is really the "take home message" of your work? I do understand that negative results are results but in a way you do not really present them as such. For example, most of your introduction praise the capabilities of two phase depth averaged model in "support the confirmation or rejection of conceptual models" stating the intrinsic epistemological potentiality with respect to one phase models. However, you then proceed with your modelling, that anyhow requires calibration and the selection of vague "physical plausible" parameters, that has numerical issues that constrain you to use "physically implausible" parameters and that do not perform well in the reconstruction of the actual phenomenon. So rather than titling your paper as "back calculation of the 2017: : :" I would suggest to switch it to something such as "challenges and open issues regarding the modelling of the 2017: : :.".

Confirmation and rejection of conceptual models is probably too much emphasized in the introduction of the discussion paper – in fact it is not the main aim of the work to find out which of the two scenarios is the "winner", but rather to investigate on how well the two scenarios can be reproduced, and what are the main challenges in doing so. The observation of the reviewer that we cannot find out the "winner" scenario while having to optimize the parameters is absolutely correct. We have reformulated the introduction and extended the discussion accordingly (confirmation and rejection of conceptual models are now treated in the discussion, L343–345):

*Confirmation or rejection of conceptual models with regard to the physical mechanisms involved in specific cases would have to be based on better constrained initial conditions, and the availability of robust parameter sets.*

The title, in contrast, is appropriate as it is, we think. The results are far from perfect, of course, but, still, most of the characteristics of the flow can be "reasonably" (see response below) reproduced, and this is only possible with particular two- or multi-phase models – aspects which are discussed in more detail in the revised manuscript (see e.g. L357–372).

These are my other comments regarding your paper

BROAD COMMENTS

1) Optimization and equifinality: The entrainment in your code is calculated with a calibrated coefficient and based on a depth averaged kinetic energy. You defined 6 zones with different friction angles and other calibrated coefficient. How did you suppose it could lead to a selection to confirm of reject a conceptual model (l 51) when, as you stated yourself in the end, there is an obvious problem of equifinality? This issue is common in back analysis, especially when several parameters are involved in the calibration process. To try to give some "physical plausibility" to the whole parametrization of the backward calculation it is important to:

i) define a straightforward and explicit optimization method ii) use parameters that are somewhat geotechnically believable with respect to the observed phenomena iii) provide a clear geomorphological/mechanical reason for each zonation – otherwise of course the more are the zones in which the parameters may be changed, the more the equifinality issue arises.

In my opinion point i) is lacking in your paper. The metrics you use are not so straightforward, especially if you need to jump between two papers to reach their definition and reason of being (is it really Mergili 2018b the best paper to refer to or is it better to go directly to Formetta 2015 and Mergili et al., 2017?). Please devote a paragraph to the interpretation of these metrics rather than cut the discussion off with "indicators of a reasonably good corrispondance". And what is reasonably anyway? Please also show in the picture a zoom of the deposition pattern (modelled and observed) in the alluvial fan. Point ii) is also important. Finding a 45 angle of friction in the E zone is rather "physically un-plausible" as it is, especially when your solid fraction decreases. You discuss this too briefly in the end of the discussion chapter. You have to explain better what is the issue with the code, is the 10 m sampling? is a projection issue related to the conversion of the coordinates? This should be better discussed, also in the light of the new titling of the manuscript. The zoning (point iii) should be more extensively discussed, the definition that it is found in table 1 is too synthetic. For each zone and limit there must be a defined reason to be it that way.

Thank you very much for this detailed and comprehensive comment. It helps us to better formulate some of the main points and challenges of the work:

- Straightforward and explicit (automated) optimization method: in principle, such methods exist in the literature (e.g. Fischer, 2013) and are available to the authors. However, they have been developed for optimizing globally defined parameters (which are constant over the entire study area) against runout length and impact area, and such tools do a very good job for exactly this purpose. However, they cannot directly deal with spatially variable parameters, as they are defined in the present work. With some modifications they might even serve for that – but the main issue is that optimization also considers shapes and maximum values of hydrograph discharges, or travel times at different places of the path. It would be a huge effort to trim optimization algorithms to this purpose, and to make them efficient enough to prevent excessive computational times – we consider this as an important task for the future which is out of scope of the present work. Therefore, we have to use a step-wise expert-based optimization strategy. This is discussed accordingly in the revised manuscript. Regarding the reference parameter set, we still think that Mergili et al. (2018b) is most appropriate, as it is newer than the other papers, therefore based on more experience and a newer version of the software, and closest to the Piz Cengalo-Bondo event in terms of process type (even though much higher in magnitude).

- Plausibility of parameters: the 45° friction angle would only apply if the flow would consist of pure solid. In the simulation, it is reduced linearly with the fluid fraction. As the fluid fraction is commonly >50% in Zone E, the friction angle of the solid is <20°. This aspect, which helps to adapt the effective characteristics of the solid depending on the fluid fraction, is made clear in the revised manuscript. Besides the explanation under the superscript 1) of Table 1, we have added the following sentence at the end of the third paragraph of Section 4 (L253-254): *It is further important to note that δ scales linearly with the solid fraction – this means that the values given in Table 1 only apply for 100% solid.*

- "Reasonably good correspondence": as there are no fixed criteria available to our knowledge what is a good correspondence, this is to some extent based on expert knowledge, summarizing the essence of Table 3. In Table 3, we have now defined the levels of empirical adequacy: empirically adequate = within the documented range of values; empirically partly adequate = less than 50% away from the documented range of values; empirically inadequate = at least 50% away from the documented range of values. The arithmetic means of minimum and maximum of each range are used for the calculation.

- We have added detail maps of the alluvial fan in Fig. 8 and Fig. 11, showing the observed and simulated impact areas.

- The zones were defined according to geomorphologic criteria and dominant process type. We have tried to formulate this in a clearer way in the revised manuscript. The newly introduced Fig. 4 further illustrates these characteristics and process types, in addition to Table 1.

2) Mass balance: in 220 you write that "only heights <0.25 m are taken into account for the visualization and evaluation of the simulation results". That's ok for the visualization part but what about mass balance? how much impact do have diffusion effects in your model? how much material you discard when you filter at 0.25 m?

This is an important point. The threshold of 0.25 m does not affect the mass balance, as it is only applied to visualization and evaluation, but not to the simulation itself. However, there is also a minimum flow height considered in the simulation – a value of 0.01 m was chosen in this case. In scenario 2, the volume of the initial landslide decreased from 3.462 million m³ at release to 3.437 million m³ at deposition, which is 0.7% and therefore negligible, at least for the purpose of the present study. Indeed, loss of material was an issue in the rather channelized Bondasca Valley, where the flow boundary, where loss of material occurs, is large compared to the flow width and flow area: in scenario 1, almost 12% of the flow material "disappeared" due to numerical diffusion until the flow reached the outlet of Val Bondasca. In the revised manuscript, we have reduced the numerical loss of material by recomputing both scenarios, decreasing the minimum flow height for the simulation from 0.01 to 0.001 m: now, the losses until the outlet is reached are below 1% for the Scenario S1, and below 4% for the Scenario S2 for each phase. This has increased the volumes reaching the outlet of Val Bondasca, but did not change the general patterns and messages. Figures and numbers were updated accordingly in the revised manuscript, and the following statement was added at L257–259:

*A threshold of 0.001 m is used for the simulation itself, keeping the loss due to numerical diffusion within a range of <1–4% until the point when the flow first leaves the area of interest.*

SPECIFIC COMMENTS

l 47-49: as a matter of taste I do not think that putting 14 references after a sentence contributes much to the clarity and readability of a paper and to the whole general usefulness for supporting a scientific discourse (that should be the main reason for inserting citations in an introduction)

The intention of the large number of references put here was to highlight the importance of the topic and to strongly support the statements. However, we can follow the argument of the reviewer and have reduced the number of references to four articles published in the last ten years.

l 88: insert a couple of words to explain how these displacements were monitored

This information was added: mainly radar interferometry, but also laser scanning was applied to measure the displacements.

l 94 and following: check that each acronym has its own definition the first time they appear in the text. Moreover the VAW and WSL reports are written in German so it is not easy to extract the required information. Please if you refer to these works add a sentence summarizing the useful findings.

We have added an introductory paragraph to Section 2.2, where the abbreviations are explained (L101-104):

*"The complex landslide which occurred on 23 August 2017 was documented mainly by reports of the Swiss Feder-al Institute for Forest, Snow and Landscape Research (WSL), the Laboratory of Hydraulics, Hydrology and Glaci-ology (VAW) of the ETH Zurich, and the Amt für Wald und Naturgefahren (Office for Forest and Natural Hazards) of the canton of Grisons."*

Those main points of these sources which are relevant for the present work are mentioned in the text, particularly in the sections 2.2 and 2.3.

l 100-102: insert data about the average steepness of the tract. In fact in the whole paper little information about the local heights and steepness are inferable. In Fig 1 the contour lines labels are missing and in the following maps the contour lines are missing altogether. I would suggest the authors to add the labels in fig. 1 and to insert a table or a figure with the average steepness of the channel profile in each of the 6 zones.

Indeed, this is an important point. In the revised manuscript we provide an additional figure (the new Fig. 4) showing a profile of the path with the elevation and slope information added, and also some information with regard to the individual zones. For this purpose, we have re-analyzed the geometric properties and included some minor updates of travel distances, drop heights, and angles of reach. Further, Fig. 1 has been equipped with contour line labels.

l 109: is it possible to talk about rock avalanche with an angle of reach of 28 and no brandung? refer to Nicoletti 1991, Corominas 1996 and Hungr 2005 and discuss

We follow the revised Varnes classification (Hungr et al., 2014) – there, rock avalanches are essentially described as pieces of rock moving as one mass like a flow, instead of individual blocks. This clearly corresponds to those descriptions of the event in Zone C which are available to us. Nevertheless, we fully agree that the issue of the angle of reach and run-up is very relevant in this context, deserving some further attention. Therefore, we have added the following text (L125-129):

*"This value is higher than the 22° predicted by the equation of Scheidegger (1973), probably due to the sharp deflection of the initial landslide. Following the concept of Nicoletti and Sorriso-Valvo (1991), the rock avalanche was characterized by channelling of the mass. Only a limited run-up was observed, probably due to the gentle horizontal curvature of the valley in that area (no orthogonal impact on the valley slope; Hewitt, 2002)."*

Corominas (1996) and Hungr et al. (2005) do not explicitly consider rock avalanches, but other types of landslides, so in our opinion it would not be appropriate to include these references here.

l 116: did you filtered the errors in the volume estimation? If yes, how?

The volume was calculated comparing the DSM made by photogrammetry and the DEM from the government with a 2 m resolution. Errors might come from the fact that we compare the 2017 DSM with a DEM previous to the 2011 event. The structures on the surface were used to exclude the 2011 volume, a procedure connected to an uncertainty of few 100,000s of cubic metres, when comparing the most plausible boundaries between the release areas of the two events. This may also explain the slight discrepancies between the volumes reported in different sources. Compared to the initial manuscript, we have revised the volume estimate and arrived at 3.2–3.3 million m³ for the initial rock slide, which corresponds better to other reports than the initial estimate of almost 3.5 million m³ (see the updated Fig. 5). No further filtering was carried out.

Fig 8 – please put hydrograph a and b to the left

We have rearranged the panes of Fig. 8 accordingly.

**Back-calculation of the 2017 Piz Cengalo-Bondo landslide cascade with r.avaflow**

*Martin Mergili, Michel Jaboyedoff, José Pullarello, Shiva P. Pudasaini*

**Response to the comments of Referee #2 (Brian McArdell)**

We would like to thank the reviewer for the constructive remarks. Below, we address each comment in full detail. Our response is written in blue colour. Changes in the manuscript are highlighted in yellow colour.

This manuscript describes the application of r.avaflow to a two-process sediment cascade involving a rock avalanche and subsequent debris flows. While the manuscript is interesting, clear and concise and is certainly of interest to users of r.avaflow, the manuscript has the character of a case study and the application of the model and concepts to other field sites is limited by the fact that the model must be calibrated for every new application. This suggests that r.avaflow is perhaps only useful for post-event analysis and not for predictive hazard analyses.

Yes, this is an important point. Indeed, we currently do not recommend r.avaflow for predictive simulations of complex landslides, as the understanding of the parameterization is not good enough. It is exactly this type of study presented here which we consider important to improve this understanding and the related challenges. We have tried to better highlight this key point of the work in the discussion and the conclusions of the revised manuscript. At the beginning of the last paragraph of the discussion, we have therefore added the following sentence (L421-424):

*The present work is seen as a further step towards a better understanding of the challenges and the parameterization concerning the integrated simulation of complex mass flows. More case studies are necessary to derive guiding parameter sets facilitating predictive simulations of such events (Mergili et al., 2018a, b).*

And at the end of the conclusions (L450-451):

*Finally, more case studies of complex mass flows have to be performed in order to derive guiding parameter sets serving for predictive simulations.*

r.avaflow is not the only model capable of describing such coupled processes, but readers may well get this impression after reading the manuscript. At a minimum, a discussion of other models and how they also described similar process transitions would be helpful for the reader, e.g. Iverson et al.'s work on runout modelling of the Oso landslide. The manuscript requires some new literature sections (other models), some new discussion sections (shortcomings and advantages of both models), and a few relatively minor clarifications (below) before it can be published. Some minor points are listed below.

See detailed comments below.

General Comments

1. Is it accurate to call r.avaflow a GIS-based model? Certainly the user interface relies extensively on a GIS interface, but I was under the impression that the model is based on a numerical solution of the shallow-water equations for granular flows, e.g. the flow model written mainly by Shiva Pudasaini.

This is a good point. In our opinion, it is an issue of wording. We would certainly call r.avaflow a GIS-based *tool*, as it strongly relies on GIS data as input, and also provides GIS data as output. The Pudasaini (2012) *model* is not directly GIS-based, we fully agree with that. Therefore, we refer to r.avaflow as a GIS-based *tool* in the revised manuscript, instead of a GIS-based numerical *model*.

2. While it is certainly efficient to refer to previous publications about the model and to not re-state the equations used in the model, I think that some description of the equations solved in the model would be appropriate for the reader. Otherwise, it is impossible to understand the manuscript without referring to the paper where the model is described. Does it include terms to describe the influence of curvature on the solution?

This is, in our opinion, a very controversial issue with this type of case studies. Even though we can follow the thoughts behind the suggestion, the problem is that just showing the main equations would open up more questions than it would answer – covering the equations in sufficient detail would make the paper extremely lengthy and would distract the readers from the actual topic of trying to reproduce the Piz Cengalo-Bondo event and push the work in a direction which differs from the intended one. Therefore, we strongly prefer to refrain from re-describing the fundamental equations behind the tool. The effects of curvature are implicitly reflected in the optimized parameter set, but are not explicitly considered in this specific set of simulations. A comprehensive and proper implementation of curvature terms will be an important aspect in the future (we are already working on it), but is not subject of the present work.

3. Section 5: Please define CSI, D2PC, and FoC. Interested readers should at least be able to see how they are defined without having to refer to the Mergili et al., 2018b paper. This is only a few sentences and it would save interested readers a fair amount of work.

Yes, this is a good point. In the revised manuscript, we have added a brief description of these three parameters at the end of the first paragraph of Section 5 (L269-273):

*CSI and D2PC measure the correspondence of the observed and simulated impact areas. Both indicators can range between 0 and 1, whereby values of CSI close to 1 and values of D2PC close to 0 point to a good correspondence. FoC indicates whether the observed impact areas are overestimated (FoC > 1), or underestimated by the simulation (FoC < 1). More details are provided by Formetta et al. (2015) and by Mergili et al. (2017, 2018a).*

4. Lines 211, 284, 290,340, and possibly elsewhere: I have no idea what you mean by the term "empirical adequacy" which sounds like it could include anything that is physically possible (instead of physically plausible). Please use a different expression.

*Empirical adequacy* just means that the simulation results are in line with the reference data, telling nothing about physical plausibility. The term was used e.g. by Oreskes et al. (1994) and, in our opinion, describes exactly what we want it to describe. We would not know about a better term for that.

Oreskes, N., Shrader-Frechette, K., & Belitz, K. (1994). Verification, validation, and confirmation of numerical models in the earth sciences. Science, 263(5147), 641-646.

5. Lines 284-292: I've heard similar statements by Prof. Florian Amann and I believe it was also mentioned by S. Demmel in her masters thesis, or in the WSL 2017 report. I have seen this idea presented this several times at different conferences, so it is unfair to not at least mention that it is not your hypothesis.

Yes, we have added the reference to Walter et al. (2019) where mentioning the injection of water into the rock avalanche (L343).

6. Others (e.g. Iverson & George) have developed similar two-phase models that are capable of describing process chains. This paper gives a false impression that r.avaflow is the only model capable of describing the runout of coupled processes. Please include a few paragraphs describing and/or comparing and contrasting your model with previous ones.

Yes, we can follow this point of criticism. In the revised manuscript, the model of Iverson and George (2014) and the software D-Claw are mentioned in the introduction, and also in the Discussion. However, in our opinion this approach shows various shortcomings, compared to the Pudasaini (2012) model: Iverson and George (2014) presented a depth-averaged two-phase mass flow model which includes five field variables: the evolution of the solid volume fraction, basal pore-fluid pressure, flow thickness and the bulk velocities. The source terms account for the influence of the granular dilation rate. The model is based on several postulates and empirical relations, including the dilation rate. However, the model assumes that the solid and fluid velocities are the same. Instead of naturally occurring six field variables, in their model, there are only five field variables. So, from the general two-phase perspective, their system is not complete, as in general, the relative velocity between the solid and fluid phase is not negligible. Moreover, their pore fluid pressure evolution equation only includes pore pressure advection and some source terms associated with dilation, but ignores the pore fluid diffusion, another potentially important aspect of fluid flow dynamics in mixture mass flows (Iverson and Denlinger, 2001; Pudasaini et al., 2005). There are no real interfacial momentum transfers, such as the drag force, virtual mass force and buoyancy between the solid and fluid phases. Furthermore, as the fluid pressure evolution is assumed to play a substantial role in their debris flow model, then the solid and fluid dynamics cannot be similar, and thus assumption of negligible relative velocity between solid and fluid is questionable (Pitman and Le, 2005; Pudasaini, 2012). Iverson and George (2014) do not consider viscous shear stress, which can play crucial role in the natural debris flows (Pitman and Le, 2005; Takahashi, 2007; Pudasaini, 2012). Moreover, in situations when the solid material behaves as isotropic material, in Iverson and George (2014) there is no dynamical coupling between the pore fluid pressure evolution and the bulk momentum equations, interactions appear only via source terms that also disappear in liquefaction.

As discussed, the Pudasaini (2012) model is the most general two-phase mass flow model that exists today, that includes all the basic mechanics and flow dynamical aspects of real two-phase solid fluid mixture mass flows with strong interfacial momentum transfer between the phases. The model can be used for any mixture composition of solid and fluid, including dry and fluid limits. So, it has been successfully applied to different flow scenarios including the cascading natural events (Mergili et al., 2017, 2018a, 2018b).

Although the Iverson and George (2014) model is restricted from the real phase-interactions, this model could still be useful for very dense debris flows where the solid particles and fluid molecules move together. The model has been solved with in an open source software, called D-Claw (George and Iverson, 2014). And the model output has been compared with some results from two sets of large-scale experiments of debris flow mobilization (Iverson RM, Reid ME, Iverson NR, LaHusen RG, Logan M, Mann JE, Brien DL 2000 Acute sensitivity of landslide rates to initial soil porosity. Science 290, 513-516. (doi:10.1126/science.290.5491.513)), and debris-flow dynamics and deposition (Iverson RM, Logan M, LaHusen RG, Berti M 2010 The perfect debris flow? aggregated results from 28 large-scale experiments. J. Geophys. Res. 115, 1-29. (doi:10.1029/2009JF001514)), both with dense debris materials.

Since there are no explicit and independent equations for the solid and fluid phase momentum transfer the model by Iverson and George (2014) cannot be applied for cascading mass flows, such as landslides impacting fluid reservoirs (Pudasaini, 2014; Kafle et al., 2016, 2019), or glacial lake outburst floods (Kattel et al., 2016).

Consequently, the model of Iverson and George (2014) is mentioned in the introduction of the revised manuscript, and the following statement (mostly summarizing what was said above) has been added to the discussion (L357-372):

*Still, we currently consider the Pudasaini (2012) model – and the extended multi-phase model (Pudasaini and Mergili, 2019) – best practice, even though other two-phase or bulk mixture models do exist. Most recently, Iverson and George (2014) presented an approach that has been solved with an open source software, called D-Claw (George and Iverson, 2014), and compared to large-scale experiments considering dense debris materials (Iverson et al., 2000; Iverson et al., 2010). The Iverson and George (2014) model can be useful for very dense debris flows where the solid particles and fluid molecules move together. However, its applicability to cascading mass flows is limited for the following reasons: (i) this model assumes that the solid and fluid velocities are the same, an assumption that does not hold for complex, cascading mass flows; (ii) the pore fluid pressure evolution equation includes pore pressure advection and source terms associated with dilation, but ignores the pore fluid diffusion; (iii) there are no real interfacial momentum transfers, such as the drag force, virtual mass force, and buoyancy between the solid and fluid phases; and (iv) neither viscous shear stress, nor dynamical coupling between the pore fluid pressure evolution and the bulk momentum equations are considered. Furthermore, as the fluid pressure evolution is assumed to play a substantial role in the Iverson and George (2014) model, the solid and fluid dynamics cannot be similar, and thus the assumption of negligible relative velocity between solid and fluid is questionable (Pitman and Le, 2005; Pudasaini, 2012).*

Specific comments

Lines 85-86: The statement that the deposit did not connect to the main channel of Val Bondasca is incorrect. The 2011 deposit certainly covered the uppermost part of the Val Bondasca and therefore is connected hydrologically and geomorphologically. The main torrent channel after the 2011 event went over and eroded the deposits of the 2011 rock avalanche deposit. Please correct this error, or more precisely state how it was "not connected". This is clearly evident in publications of others, e.g. Frank et al. (2019) which illustrates the triggering of debris flows in Val Bondasca following the 2011 rock avalanche. Frank, F., Huggel, C., McArdell, B. W., & Vieli, A.(2019). Landslides and increased debris-flow activity: a systematic comparison of six catchments in Switzerland. Earth Surface Processes and Landforms, 44(3), 699-712.https://doi.org/10.1002/esp.4524.

*Thanks a lot for this comment and the reference. We have changed the text accordingly (L89-93):*

*On 27 December 2011, a rock avalanche with a volume of 1.5–2 million m³ developed out of a rock toppling from the NE face of Piz Cengalo, travelling for a distance of 1.5 km down to the uppermost part of the Val Bondasca (Haeberli et al., 2013; De Blasio and Crosta, 2016; Amann et al., 2018). This rock avalanche reached the main torrent channel. Erosion of the deposit thereafter resulted in increased debris flow activity (Frank et al., 2019).*

Lines 102-103. These data are almost certainly from the Canton of Grisons and not from the WSL, although they were certainly repeated in the WSL 2017 report.

*We only found the data in the WSL report and have completed the reference as follows:*

*data from the Canton of Grisons reported by WSL, 2017*

Lines 129-130. This statement about the pore-water pressure rise is somewhat speculative and should be treated as a hypothesis and not as a fact. Rock avalanche deposits (e.g. the 2011 deposits) are often described as having relatively low porosity, and therefore one could, possibly, also expect dilation upon compression which would cause a drop in pore-water pressure and not a rise. The river-bed sediments, and possibly the moraine deposits, upon which the 2017 rock avalanche flowed would be more reasonable candidates for the mechanism you describe here.

*We absolutely agree that the pore water pressure rise is a hypothesis rather than a fact. This is also clearly indicated by the start of the sentence: Walter et al. (2019) claim that …*

Lines 221-222: Did you try other grid resolutions? This resolution, based on my own experience using CFD models (also with a GIS user interface) indicates to me that the grid size is fine for the rock avalanche runout modelling but too coarse for the debris-flow runout modelling.

*This is an important issue, indeed. We have also tried 5 m resolution, but the results for this specific case were not substantially different, only the computational time increased considerably.*

Line 237: What do you mean by total discharge? The sum of the discharge of all of the phases, or the sum of different flow paths comprising the flow. Or do you mean maximum discharge or the maximum discharge sum of all of the phases? From the text is appears to be a peak discharge value and not just a total value.

*With total discharge, we meant solid+fluid discharge. The text was reformulated accordingly in order to avoid confusion.*

Line 346: A brief re-statement of the Scenarios would be reasonable here for the reader.

We have reformulated the start of the conclusions as follows (L436-437):

*Both of the investigated Scenarios S1 (debris flow developing at the front of the rock avalanche) and S2 (debris flow developing at the back of the rock avalanche, overtopping the deposit) …*

Table 1: It's confusing to have Zones A, B, and so on, as well as scenario sets A & B. perhaps you could use lower-case letters for scenarios a and b, or use something like SA or SB to make it completely unambiguous.

Yes, this is true – we have changed the names of the scenarios to S1 and S2.

Figure 5 is a nice graphical depiction of the two scenarios.

Thank You!

[revised manuscript text omitted]

---

## Author Response (AR2)

**Back-calculation of the 2017 Piz Cengalo-Bondo landslide cascade with r.avaflow**

*Martin Mergili, Michel Jaboyedoff, José Pullarello, Shiva P. Pudasaini*

**Response to the comments of Referee #1**

We would like to thank the reviewer for the constructive remarks. Below, we address each comment in full detail. Our response is written in blue colour. Changes in the manuscript are highlighted in yellow colour.

Dear authors. I appreciate your answer that has addressed all of my comments. I think that the paper now is clearer and that in this form is somewhat more interesting for the readers (a case study is always just a case study, the narration of the struggling with the modelling of a case study, with all the problems well addressed, is more interesting). I still have some comments, but they are minor ones.

**Broad comments:**

1) I still have some issues with the title, consider to incorporate "key challenges" – a form you use yourself in line 73 – in the title

We understand this point and have modified the title as follows: ***Back-calculation of the 2017 Piz Cengalo-Bondo landslide cascade with r.avaflow: what we can do and what we can learn***

2) Even though it is good that you explicit the fact that $\delta$ scales linearly with the solid fraction, you would agree with me that it is somewhat strange that the solid particles of your flow increase their friction angle along their path. Probably you also increase the fluid fraction and the "real, combined, effective $\delta$" (the equivalent of a single phase model) is lower but if so I still believe that this should be explicated in the table with another column (either solid fraction or "effective $\delta$"). Moreover, if in zone E the flow is so channelized how comes that no entrainment occurred (lateral instabilities for examples) when such a mass of solid and fluid material flowed in its path? Are zone D and E all composed by rocky outcrops (it does not seem so from the pictures)?

The reviewer raises an important issue here, which we have now clarified in the discussion (L427-433):

*"The higher values of $\delta$ in the lower portion of the channel are based on the assumption that $\delta$ of the solid material would somehow depend on the momentum or energy of the flow, which – due to the relatively low velocity – is much less in the zones D and, particularly, E. While this assumption, in our opinion, is justified by fluidization and lubrication effects often observed – or inferred – for very rapid mass flows, it remains hard to consider those effects by a well-justified numerical relationship. Until such a relationship (which definitely remains an important subject of future work) has been proposed, we rely on empirically-based zonations of friction parameters."*

Providing values of "effective $\delta$" in the table would be hard as the solid fraction changes dynamically.

Regarding the entrainment: indeed, the narrow gorge in zone E is incised in a bedrock swell (the outlet of the hanging valley of Pleistocene origin). In general, flow velocities and applied shear stresses are low in the zones D and, particularly, E, so that the assumption of no entrainment appears plausible.

3) Table 3. The threshold of 50% for empirical adequacy allows clearly a large space of wiggle room and it is maybe too generous. However the importance is that it is explicated in the text so the reader could weight better the results

Yes, we agree that the 50% are a rather "generous" margin. However, it has to be kept in mind that also the observations are uncertain, so that too narrow margins would be questionable. We have added the following sentence to the discussion (L439-440):

*"Also, the other reference data are not exact. Therefore, we allow a broad margin (50% deviation from the observation) for considering the model outcomes as empirically adequate."*

**Specific comments**

1) Line 57-61: this sentence is not clear and well articulated

Yes, we understand this point of criticism. Besides slightly changing the wording, we have converted the running text to a list, which hopefully improves the ease of reading (L59-68):

(i)     *model cascades, changing from one approach to the next at each process boundary (Schneider et al., 2014; Somos-Valenzuela et al., 2016). Each individual model is tailored for the corresponding process component;*

(ii)    *bulk mixture models or two- or even multi-phase flow models (Pitman and Le, 2005; Pudasaini, 2012; Iverson and George, 2014; Mergili et al., 2017; Pudasaini and Mergili, 2019). Two- or multi-phase flow models separately consider the solid and the fluid phase, but also phase interactions, and therefore allow to consider more complex process interactions such as the impact of a landslide on a lake or reservoir.*

*Worni et al. (2014) have highlighted the advantages of (ii) for considering also the process interactions and boundaries.*

2) Line 149: approximately

Thank You, corrected!

3) Line 387: please explicit what these "numerical issues related to the narrow gorge" are

We have clarified this issue as follows (L397-402):

*"… the steep walls of the gorge, in combination with the low number of raster cells representing the width of the flow, challenge the correct geometric representation of the flow in the topography-following coordinate system. Further, application of the NOC-TVD scheme results in numerical diffusion which becomes particularly evident in this situation. The introduction of adaptive meshes – which would help to locally increase the spatial resolution while maintaining the computational efficiency – could alleviate this type of issue in the future."*

**Back-calculation of the 2017 Piz Cengalo-Bondo landslide cascade with r.avaflow**

*Martin Mergili, Michel Jaboyedoff, José Pullarello, Shiva P. Pudasaini*

**Response to the comments of Referee #2 (Brian Mc Ardell)**

We would like to thank the reviewer for the constructive remarks. Below, we address each comment in full detail. Our response is written in blue colour. Changes in the manuscript are highlighted in yellow colour.

The manuscript has been substantially improved from the last version. It describes the difficulties in modelling the Piz Cengalo-Bondo mass movement event cascade. I still have a few questions about the manuscript that I think should be considered before publication. I still think that the manuscript will be of interest to the readers of NHESS.

Lines 75-79: I'm not sure what the policy of NHESS is regarding repetition and word count, but lines 75—79 don't really add much to what has already been written in the previous page and therefore could be omitted. Instead, I would prefer more test to explain some novel aspects of your model, e.g. the nature of the coupling between the phases (another comment on that below).

We agree that there is not much new information in this paragraph. Its purpose is to lead over to the following sections and to facilitate following the storyline. We would therefore rather prefer to keep this piece of text.

Line 134: What is MNS? I presume that it's the French-language abbreviation of the English term DSM? Most readers won't be familiar with the French abbreviation. Additionally, why did you use the surface model from Swisstopo instead of the terrain model? The surface model would also include vegetation, so using a surface model from 2011 would have included about 6 years of vegetation growth for areas not modified by the 2011 event, and vegetation removed by the 2017 event would lead to systematic errors in your mass balance. Many trees which survived the 2011 rock avalanche were destroyed by the 2017 event, so this could easily lead to errors. I presume that this has been accounted for by the authors, however it may not be clear to readers.

MNS means "modèle numérique de surface" in French. Indeed, it is the equivalent term to "digital surface model".  DTM (en) = MNT (fr) and DSM (en) = MNS (fr). The term DEM may be used for different things, therefore we avoid this term in the revised manuscript and have replaced it by *"DTM"* throughout the manuscript. MNS was incorrect, MNT would have been correct, but we have removed the French terms anyway.

We compared the source area from the DTM (swisstopo, 2011) to the model we created by structure from motion (DSM, 2017) to assess the volume. At the top area the vegetation is not a problem, but the snow/glacier change between 2011 and 2017 affects the volume (this aspect is discussed at L143-144). The elevation map for the propagation modelling is the DTM from 2011, and not the DSM from 2017. The reason is, as highlighted by the reviewer, the vegetation, noise and artefacts in the DSM.

Section 3: Please include more details on how the fluid and solid phases are coupled. You gave carefully-worded explanations in the response to the reviews, and it would certainly be interesting for the reader to see the details of how the phases are coupled—even if only in plain text without equations. When looking at the video of the rock avalanche event and the videos of the various debris-flow surges in the village of Bondo, it is clear that phase separation is important, and this manuscript would be incomplete without at least some description of the coupling, as well as some description of the difference in travel speed of the two phases.

Yes, we agree with this suggestion and have added the following text to the section about r.avaflow (L203-207):

*"The solid and fluid phases have their own mass and momentum balance equations, so that they evolve as independent dynamical quantities while the phases are still coupled. This means that, in general, the solid and fluid velocities are different. However, the use of an enhanced drag model (Pudasaini, 2019) and the consideration of virtual mass forces ensure a strong coupling between the solid and the fluid phases in the mixture (Pudasaini, 2012; Pudasaini and Mergili, 2019)."*

Line 238 ff: The model has been tuned by adjusting the coefficients in six different regions of the runout zone. This strongly suggests that the model cannot be used for prediction without calibration. To help the reader understand how important the parameter tuning exercise is for applying the model, perhaps you could compare the results with results calibrated based only on an inspection of the aerial photographs, maps, and other materials prior to the 2017 event.

This is a very interesting idea, but we think that such an analysis could hardly be done at this point, particularly not within the scope of this publication. We would like to highlight two aspects:

- With the results of the calibration procedure with the 2017 event already available, it would be almost impossible to do such a parameterization in an unbiased way.

- The results would very much suffer from subjectivity of the parameterization. However, it could be a really useful future study to let a couple of scientists parameterize the model independently, based on pre-event data only (for this or another event), and then analyze the difference in the outcomes. However, this would be a separate study and would go beyond the scope of the present paper.

Section 5.1: Are the statistical metrics CSI and D2PC from Formetta et al., 2015? I haven't heard of these metrics before, or perhaps not with these names. This section is a bit confusing to read because the meaning of the statistical metrics and critical values of those parameters is only roughly explained.

These metrics have previously been used for slope stability analyses, before we introduced them into r.avaflow. The metrics are explained in detail in Formetta et al. (2015), Mergili et al. (2017) and also Mergili et al. (2018a), so that it does not appear necessary to reiterate everything in detail also in this publication.

Line 283: What is the justification of using a fluid with a density of 1400 kg m^-3? Is this an empirical value or based on some physical assumptions? This value seems to be quite large, and appears to have been arbitrarily defined. It is large enough that your solid concentrations will consequently appear to be quite small.

Fluid, in this case, is not pure water, but mud, i.e. a mixture of water and solid. In this case, we assume approx. 75% water (1000 kg/m³) and 25% fine material (2700 kg/m³) in the mud. This is a reasonable assumption from our point of view.

Line 328: Please explain what you mean by "leaks" or re-write the sentence to more clearly explain the effect.

We have replaced "*leaks*" with "*releases material*".

Line 357 & 358. Please remove the sentence about this model being considered "best practice." Given only one case study, with all of the tuning that went into reproducing the observations, and the other limitations described in the preceding paragraphs, this statement cannot be really taken seriously.

We do not agree with this statement, as it has to be distinguished between the model and the data. As is explained in the text of the revised manuscript, there is no other operational software available offering a comparable physical basis, and this is we refer to with that sentence. On the other hand, there is the parameterization which – in this point we completely agree – are uncertain, and we need more case studies like this one to build a set of guiding parameter values comparable to the one which exists, for example, for RAMMS, for "simpler" types of processes. This limitation is clearly stated in the discussion.

Line 362: Given what I've read of the Iverson & George model, it appears to also be useful for geophysical flows that would be comparable to the ones that you describe in the paper. I think that this statement in the paper is incorrect. I encourage you to focus on the advantages of your own model instead of highlighting possibly fictitious disadvantages of other models.

Based on the discussed literature, we have reformulated the corresponding part of the discussion accordingly (L375-381):

*"However, the Pudasaini (2012) model is better suited for the simulation of cascading mass flows for the following reasons: (i) solid and fluid velocities are considered separately which is important for complex, cascading mass flows; (ii) pore fluid diffusion is included, whereas the model of Iverson and George (2014) is limited to pore pressure advection and source terms associated with dilation; (iii) interfacial momentum transfers, such as the drag force, virtual mass force, and buoyancy between the solid and fluid phases are fully included; and (iv) viscous shear stress and dynamical coupling between the pore fluid pressure evolution and the bulk momentum equations are considered."*

Line 400: The sediment leaving the area was of major interest to the communities downstream. This is part of the process cascade that would be relevant for hazard managers. If your model is capable of describing sediment transport (suspended and bedlam), then you may want to highlight that here but state that you did not focus on that aspect of the work.

We agree that also the lower part of the process would be interesting for hazard managers, but the focus of our work was the upper part. We have added the following sentence (L415-416): *"That lower part of the process chain was not subject of the present work."*

Conclusions: This section is difficult to read: you may also want to include some more information about the case that you modelled in the paper and why: the conclusions should also contain a concise summary of the problem, the methods, results, and major discussions points.

This, in fact, is very true: for some reason, we have lost the beginning of the conclusions section. In the revised manuscript, we have added the following paragraph at the start of the conclusions (L456-462):

*"We have back-calculated the 2017 Piz Cengalo-Bondo landslide cascade in Switzerland, where an initial rock slide-rock fall of approximately 3 million m³ entrained a glacier, continued as a rock avalanche, and finally converted into a series of debris flows reaching the village of Bondo at a total distance of 6.5 km. The water causing the transformation into a debris flow might have originated from entrained glacier ice or from water injected from the debris beneath the rock avalanche. Considering the event from its initiation to the first debris flow surge, we have evaluated the possibilities, but also the challenges in the simulation of such complex landslide events, employing the two-phase model of the software r.avaflow."*

Table 1, Zone c: The exponent on the CE parameter is typeset as "-8-0". Is this a typo?

Yes, thank You for reading so carefully. We have corrected this mistake; the correct exponent is -8.0.

[revised manuscript text omitted]